# LLaSO: A Reproducible Foundation for Large Speech-Language Models

## Abstract

The development of Large Speech-Language Models (LSLMs) has been limited by fragmented architectures and poor transparency, making reproducibility and fair comparison difficult. In contrast to the vision–language domain, where open resources have driven rapid progress, LSLMs are often released only as model weights without their training data or configurations, leaving the field without common baselines. We present LLaSO, the first fully open, end-to-end framework for large-scale speech–language modeling. LLaSO consists of three key components: (1) LLaSO-Align, a 12M-instance speech–text alignment corpus; (2) LLaSO-Instruct, a 13.5M-instance multi-task instruction-tuning dataset for speech–text understanding; and (3) LLaSO-Eval, a standardized, reproducible benchmark for cross-modal evaluation. To demonstrate its utility, we train LLaSO-Base, a 3.8B-parameter reference model built entirely on public data. LLaSO-Base achieves a normalized score of 0.72, outperforming comparable models and providing a strong, reproducible baseline. Our analysis further shows that while broader training coverage improves performance, significant generalization gaps remain, especially in speech-only scenarios. By releasing datasets, benchmarks, and models together, LLaSO establishes an open standard for LSLMs, enabling unified research and faster community progress.

## 1 Introduction

The remarkable success of Large Language Models (LLMs) has established a powerful foundation for multimodal AI OpenAI (2024); Yang et al. (2025). In the visual domain, this has led to the rapid maturation of Large Vision-Language Models (LVLMs), where established paradigms, such as leveraging CLIP-style encoders Radford et al. (2021), have enabled effective and scalable alignment between vision and text Awadalla et al. (2023); Wang et al. (2024); Bai et al. (2025); Cocchi et al. (2025). In contrast, the development of Large Speech-Language Models (LSLMs) remains in a more nascent and fragmented stage. The field currently lacks consensus on fundamental architectural principles, with competing approaches that include external feature fusion Radford et al. (2022); Li et al. (2023b), dedicated cross-modal attention mechanisms Kong et al. (2024b); Elizalde et al. (2024), and implicit alignment strategies Chu et al. (2024).

This architectural divergence is compounded by a lack of transparency in existing research. While several open-source LSLM initiatives have emerged Chu et al. (2023); Défossez et al. (2024); Tang et al. (2024), many are only partially open. Model weights may be released, but the underlying training data and crucial configurations are often withheld. This opacity makes it difficult to conduct fair comparisons, as performance differences can be attributed as much to proprietary data or undisclosed training strategies as to architectural merit, hindering systematic progress.

To address these challenges of fragmentation and opacity, we introduce LLaSO: a fully open, end-to-end framework designed to establish foundational standards for LSLM research. LLaSO consists of three core, publicly available components:

1. **LLaSO-Align:** A 12M-instance speech-text alignment corpus aggregated from diverse sources, including conversational speech Chen et al. (2021), read narratives Panayotov et al. (2015), audio books Ito & Johnson (2017); Pratap et al. (2020), and accented speech Veaux et al. (2016).

2. **LLaSO-Instruct:** A 13.5M-instance instruction-tuning dataset covering 20 tasks across linguistic, semantic, and paralinguistic domains. It supports three distinct modality configurations: audio instructions with audio inputs, textual instructions with audio inputs, and audio instructions with textual inputs.

3. **LLaSO-Eval:** A reproducible benchmark of 15,044 stratified samples designed for comprehensive evaluation of instruction-following capabilities of LSLMs.

To validate our framework and provide the community with a strong, reproducible baseline, we developed LLaSO-Base, a 3.8B-parameter reference model that adapts the successful LLaVA architecture to the speech domain. Trained exclusively on LLaSO-Align and LLaSO-Instruct, and evaluated on LLaSO-Eval, our model achieves a normalized score of 0.72, outperforming the next best comparable model (0.65). As illustrated in Figures 1 (Middle and Right), LLaSO-Base is designed not for state-of-the-art performance, but to demonstrate the power of an open, extensible, and reproducible workflow.

Our evaluation shows that while broader training improves overall performance, models still struggle with generalization, leaving substantial gaps on unseen tasks and pure audio settings. Investigating potential solutions for this weakness, we found that models equipped with interleaving and parallel decoding mechanisms exhibit far greater robustness in these challenging scenarios.

In summary, LLaSO provides the first fully open, end-to-end stack for LSLM research, comprising large-scale training datasets, a standardized benchmark, and a reference model. By releasing these resources, we aim to lower the barrier to entry and foster a new wave of systematic, community-driven progress in large-scale speech-language modeling.

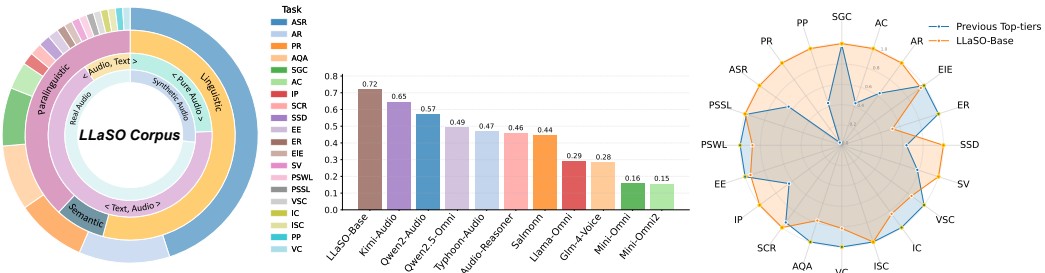

Figure 1: (Left) LLaSO Corpus Overview: 25.5M audio-text pairs over 20 tasks (18 paralinguistic), integrating LLaSO-Align, LLaSO-Instruct, and LLaSO-Eval, with 73% real and 27% synthetic audio (further statistics are detailed in Appendix Q and R). (Middle) Overall model performance after min-max normalization for direct comparison where higher bars indicate better overall performance. (Right) Normalized task-level results on LLaSO-Eval: LLaSO-Base (orange) vs. leading baselines (blue) across 20 tasks, with scores scaled by setting the best model to 1 (detailed results are provided in Appendix S).

## 2 RELATED WORK

**Vision-Language Modeling.** Vision-language modeling has rapidly advanced through a standardized two-stage paradigm: modality alignment followed by instruction tuning Brown et al. (2020); Bommasani et al. (2021); Li et al. (2023c). The rapid progress in this field has been facilitated by two essential types of open resources. First, public training datasets and standardized evaluation benchmarks Ma et al. (2023); Hsieh et al. (2023); Zeng et al. (2024b); Fu et al. (2024); Huang et al. (2025) have become widely adopted, enabling fair comparison and transparent reproducibility across models and tasks. Second, open-source implementations with modular codebases such as LLaVA Lin et al. (2023) and OpenFlamingo Awadalla et al. (2023) have significantly lowered the technical barriers to development and fostered rapid iteration across the community. Together, these practices have fostered a shared research infrastructure where new models and tasks are often built upon existing resources Liu et al. (2023); Yin et al. (2024). This has allowed vision-language research to focus more on advancing scientific capabilities rather than reimplementing foundational components.

**Speech-Language Modeling.** Compared to vision-language modeling, progress in speech-language systems has been less cohesive Su et al. (2025); Ma et al. (2024). First, most leading models such as Audio Flamingo Kong et al. (2024a); Ghosh et al. (2025), Qwen-Audio Chu et al.

(2023; 2024), and Kimi-Audio KimiTeam et al. (2025) rely on proprietary data, limiting reproducibility Peng et al. (2025); Pandey et al. (2025). Second, most models support only narrow modality configurations (e.g., text-plus-audio), with few addressing more compositional tasks Tang et al. (2024); Chu et al. (2024); Chen et al. (2024). Third, existing datasets largely focus on semantic reasoning Fang et al. (2025); Wu et al. (2024); Mei et al. (2024), with limited coverage of prosody and emotion. Lastly, few open-source stacks unify models, datasets, and benchmarks; most systems (e.g., LauraGPT Du et al. (2023), Moshi Défossez et al. (2024), Westlake-Omni Xinchen-ai (2024)) lack full releases, hindering reproducibility and community development.

| Name | Alignment Data | Alignment Tasks | Task Coverage | Modality Coverage | Audio Type | Sample Num. | Duration (Hours) |
|---|---|---|---|---|---|---|---|
| AVQA | ✗ | - | 1 | ① | Collected | ~57.3K | - |
| COTA | ✗ | - | 5 | ① | Mixed | ~1.2M | - |
| OpenAQA | ✓ | Multiple | 4 | ① | Collected | ~5.0M | - |
| OpenASQA | ✓ | Single | 8 | ① | Collected | ~9.6M | - |
| SIFT-50M | ✓ | Multiple | 10 | ① | Collected | ~55.6M | - |
| SALMONN | ✓ | Multiple | 14 | ② | Collected | ~2.3M | ~4.4K |
| **LLaSO Corpus** | ✓ | Single | 20 | ③ | Mixed | ~25.5M | ~89.5K |

Table 1: Comparison of public speech-language datasets and LLaSO Corpus. For "Modality Coverage," ① means only text instruction with audio input, ② adds pure audio formats, and ③ indicates full support, including audio instruction with text input. "Audio Type" denotes real ("Collected"), synthetic, or mixed data. ✓/✗ show whether alignment data are presented.

## 3 *LLaSO* CORPUS

To support the development of LSLMs, we introduce the *LLaSO Corpus*, a comprehensive, modular benchmark suite.

### 3.1 CORPUS OVERVIEW

Inspired by practices in LVLMs, LLaSO comprises three tightly integrated components:

- *LLaSO-Align*: A large-scale corpus for aligning speech with semantic space through ASR-based supervision.
- *LLaSO-Instruct*: A multi-task instruction-tuning dataset spanning linguistic, semantic, and paralinguistic tasks.
- *LLaSO-Eval*: A stratified benchmark designed for consistent evaluation across tasks.

These components together support the full training pipeline of LSLMs, modality alignment, instruction tuning, and evaluation (see Figure 1 (Left)).

To advance LSLMs beyond vision-language paradigms, we anchor our benchmark design in two core properties of speech:

- *Inherent Paralinguistics*: Speech conveys rich, essential information beyond words such as speaker identity, accent, emotion, and prosody. These paralinguistic cues are omnipresent and crucial for natural human communication.
- *Flexible Modality Roles*: In LSLMs, both audio and text can serve as inputs or instructions, enabling diverse interaction patterns e.g., audio-instruction with text input, text-instruction with audio input, or audio-instruction with audio input.

To better reflect the needs of real-world systems, we adopt a balanced task weighting approach that corrects for limitations in existing corpora:

- *Semantic Tasks (8%)*: Intentionally underweighted, as their success often reflects language modeling capacity rather than speech understanding Rouditchenko et al. (2025), and they are already well-represented Gong et al. (2023b); Fang et al. (2025).
- *Paralinguistic Tasks (40%)*: Prioritized to address their underrepresentation in current resources Jiang et al. (2025); yu Huang et al. (2024). We ensure diversity by combining real-world metadata with synthetically generated variations.

- *Linguistic Tasks (52%)*: Dominated by ASR, which remains foundational to grounding speech in linguistic structure and is critical for general performance.

The final LLaSO Corpus includes 71% real-world audio and 29% synthetic speech, and covers a broad range of modality configurations where both audio and text flexibly act as inputs and instructions presented in Table 1. This design ensures robust coverage of the full speech landscape, supporting the development of unified and adaptable LSLMs.

## 3.2 LLaSO-ALIGN

To establish a robust semantic foundation for speech-language modeling, we adopt ASR as the core alignment task in LLaSO Corpus's first stage. Following vision-language best practices, this approach grounds speech representations directly in textual semantic space through explicit instruction-response pairing. *LLaSO-Align* contains 12M instruction-formatted ASR samples, each including an audio input, a natural language instruction, and a reference transcript. Unlike traditional ASR datasets offering only raw audio-text pairs, we introduce 18 hand-crafted instruction templates that frame the task with varying specificity and constraints, e.g., "Transcribe the audio precisely; return text only", encouraging instruction adherence and realistic use. To ensure diversity in content and speaker profiles in LLaSO-Align, we aggregate wide-range public ASR corpora spanning conversational speech Chen et al. (2021), single-speaker narration Panayotov et al. (2015), audio books Ito & Johnson (2017); Pratap et al. (2020), and accented English Veaux et al. (2016), capturing a broad range of acoustic environments, accents, ages, and speech styles. All samples undergo a construction pipeline to ensure consistency and quality presented as Figure 2 where its standardization details in Appendix Q. By reframing ASR as an instruction-following alignment task and curating a diverse, high-quality dataset, LLaSO-Align lays the groundwork for downstream speech-language understanding across modalities.

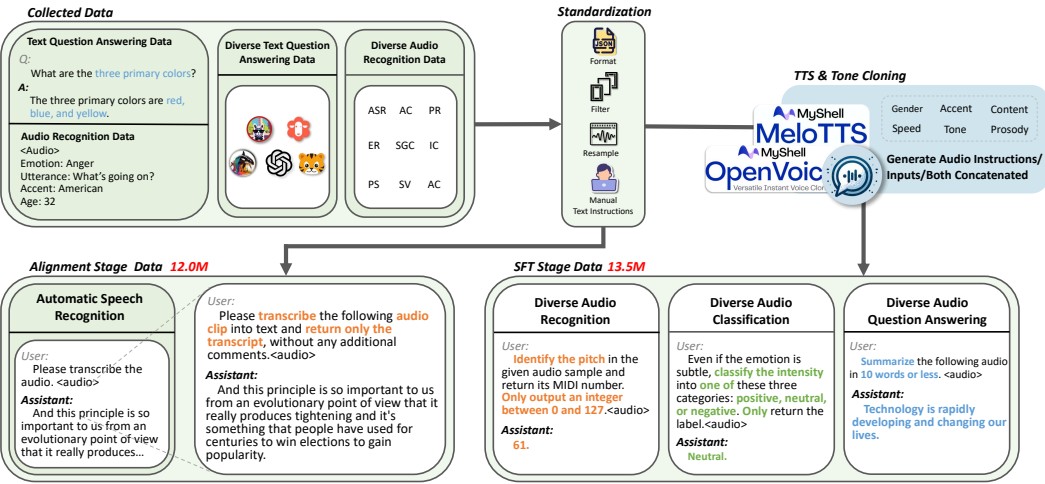

Figure 2: LLaSO Corpus construction pipeline. We first aggregate heterogeneous sources, including text-based QA corpora and speech datasets covering acoustic, paralinguistic, and semantic tasks, then normalize format, sample rate, and instruction style, etc. We construct LLaSO-Align (12.0 M) for aligning speech and text modality via ASR, while LLaSO-Instruct (13.5 M) for multi-task instruction tuning including classification, recognition, and AQA. When synthesize audio, we use advanced audio synthesis as described in Appendix H for richer speaker variation, enabling *pure-audio*, *text plus audio*, and *audio plus text* formatted samples with diverse gender, accent, speed, and tone.

## 3.3 LLaSO-INSTRUCT

Building on the aligned speech-text representations from LLaSO-Align, we present *LLaSO-Instruct*, a multi-task instruction tuning dataset designed to advance speech-language modeling with greater task diversity and richer modality configurations. Unlike previous instruction datasets focused primarily on semantic tasks with limited input modalities, LLaSO-Instruct fully embraces the inherently multimodal and paralinguistic nature of speech, systematically expanding both task coverage and modality pairings, offering a comprehensive framework to instruction tuning LSLMs.

**Task Coverage.** LLaSO-Instruct spans **20 tasks** across linguistic, semantic, and paralinguistic categories. While linguistic tasks (e.g., ASR) and semantic tasks (e.g., audio-based QA) cover foundational capabilities, the majority of tasks are paralinguistic, including speaker-centric tasks and content-centric tasks, designed to capture speaker traits and contextual acoustic crucial for socially-aware interaction, with all included tasks presented in Appendix Q. To construct this wide range of tasks, firstly, we collect task-specific datasets with rich metadata, enabling reuse of the same audio sample across multiple tasks, with its associated labels such as accent and gender. When label distributions are imbalanced, we implement targeted sampling strategies. [1] Secondly, for each task, we manually construct 20 text instructions across four prompt styles including standardized, contextualized, stylistic, and fine-grained (examples in Appendix L). For ASR and AQA tasks, instructions are open-ended inviting free-form responses from the model, while paralinguistic tasks predominantly employ closed-ended instructions, requiring the model to select an answer from predefined options without additional analysis. To address diverse task requirements, we construct training samples at multiple levels of granularity, so that some paralinguistic tasks also include open-ended variants. [2] We display task prototypes in Figure 3 and the open/closed-ended mapping in Appendix Q. As to linguistic and semantic task categories, we will discuss in the following paragraph.

**Modality Coverage.** To reflect speech's flexible role as both input and instruction, LLaSO-Instruct supports three core modality configurations: **1)** Text instruction with audio input; **2)** Audio instruction with text input; **3)** Pure audio for both instruction and input. We provide three examples in Figure 4 and an overview in Table 11. Linguistic tasks retain their native modality pairings, with one million ASR samples carried over from LLaSO-Align. Semantic QA tasks are derived from high-quality text datasets (e.g., OpenOrca and Alpaca) and converted into multimodal samples using audio synthesis (see Appendix H), thus each instance may yield multiple variants (e.g., text-with-audio, audio-with-text) to support cross-modal learning. [3] Similarly, paralinguistic tasks are primarily configured as text-instruction with audio input, but where feasible, we also construct fully speech-driven formats by synthesizing both instruction and input as audio, enabling training in pure audio scenarios better simulating human-human interaction.

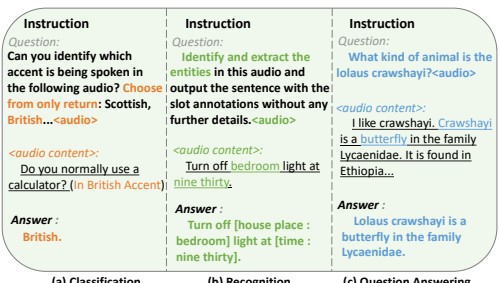

Figure 3: **Task prototypes.** (a) closed-set *classification*; (b) multi-granularity/open-set *recognition*; (c) open-ended *AQA*.

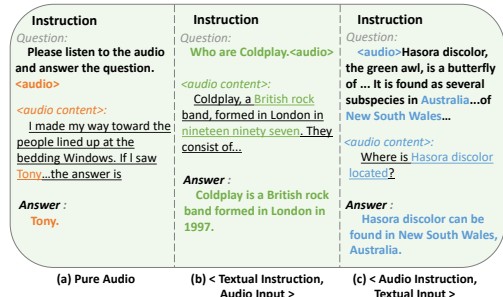

Figure 4: **Interaction formats** in LLaSO Corpus.

### 3.4 LLaSO-Eval

To complete our data trio, we introduce LLaSO-Eval, a held-out, training-disjoint evaluation suite designed to accompany the LLaSO training set. Derived from the same underlying corpus but separate from the training split, LLaSO-Eval covers 15,044 samples across 20 tasks, categorized into linguistic, semantic, and paralinguistic categories and supports all three modality configurations to test both within- and cross-modal generalization. The suite includes open-ended prompts for free-form comprehension/reasoning and closed-ended prompts enabling quantify instruction following capability via abstention rate. A task-level breakdown is provided in Appendix Q and R.

---

[1] For example, in the Meld accent dataset, we address the long-tail by removing rare accents and trimming dominant ones. Similarly, we repurpose VCTK's gender metadata for speaker classification, balancing the dataset by downsampling the female subset to 1:1 ratio.

[2] For example, in age classification, we use three levels: coarse-grained (e.g., "twenties", "fifties"), medium-grained (e.g., "15-19", "20-24"), and fine-grained where the model is required to predict the exact age as an integer between 18 and 80.

[3] When instruction and input segments are suitable for audio synthesis (English-only, properly normalized, and error-free), each textual QA instance yields both text-with-audio and audio-with-text variants.

## 4 MODEL

To validate the effectiveness of our LLaSO Corpus, we introduce LLaSO-Base, a reference model in the speech-language domain that strictly aligns with the end-to-end instruction tuning paradigm established in vision-language research Zhu et al. (2023); Cocchi et al. (2025); Li et al. (2023a). Rather than pursuing new SOTA results, our objective is to offer the community a robust and extensible baseline for systematic cross-modal instruction following.

### 4.1 MODEL ARCHITECTURE

Our model follows a simple yet proven three-component design as illustrated in Figure 5 that uniformly supports text with audio, audio-only, and audio plus text inputs via embedding concatenation. For audio features, we use the Whisper-large-v3 encoder Radford et al. (2022); Zhang et al. (2024); Gong et al. (2023a), retaining only the encoder (~640M) to leverage its strong representations while leaving generation to the LLM. Audio is processed by Whisper's front end (16 kHz log-mel, stride-2 ≈40 ms/frame) with SpecAugment Park et al. (2019) during training. Final-layer features $Z^a = F_{ae}(X^a)$ (where $X^a$ may denote audio instructions, content, or both) are projected into the LLM embedding space by a two-layer multi-layer perceptron (MLP) with Gaussian Error Linear Unit (GELU) activation, $H^a = F_{proj}(Z^a)$, chosen for its simplicity and effectiveness over heavier alignment modules Tang et al.

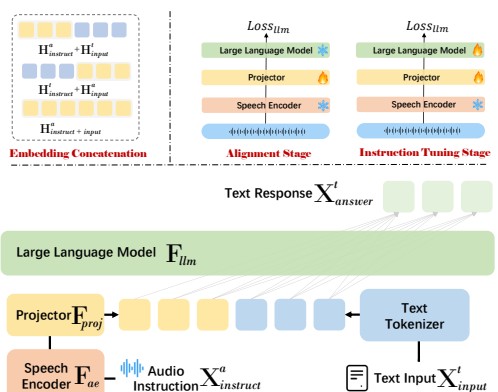

Figure 5: Overview of LLaSO-Base: model architecture and input flow (Bottom), three input layouts (Top Left), and the two-stage training recipe of alignment and instruction tuning (Top Right).

(2024); Kong et al. (2024b); Lin et al. (2024). The projected $H^a$ is concatenated with text embeddings $H^t_{instruct/input}$ from the tokenizer, yielding a unified multimodal sequence with preserving order. The sequence is subsequently processed by Llama-3.2-3B-Instruct Grattafiori et al. (2024), a mainstream instructed backbone. With ~3.8B total parameters, LLaSO-Base balances computational efficiency and representational capacity.

### 4.2 TRAINING

We train the model in a single-turn dialogue setting, where each instance consists of audio $X^a$, its paired text $X^t$, and the target response $X^t_{answer}$. To support different modality configurations, we unify the query format as in Eq. 1.

$$X^{(t,a)}_{query} = [X^t_{instruct}, X^a_{input}], \quad X^{(a,t)}_{query} = [X^a_{instruct}, X^t_{input}], \quad X^{(a)}_{query} = [X^a_{instruct+input}] \quad (1)$$

Training optimizes parameters $\theta$ via next-token autoregressive prediction, maximizing the conditional likelihood of the response given the query, as defined in Eq. 2. We adopt a proven two-stage instruction-tuning paradigm, alignment followed by instruction tuning, with the set of trainable parameters $\theta$ varying by stage.

$$p\Big(X^t_{answer} \mid X^{(*)}_{query}\Big) = \prod_{i=1}^{L} p_\theta\Big(x^t_i \mid X^{(*)}_{query,<i}, X^t_{answer,<i}\Big), \quad (2)$$

where $X^t_{answer}$ denotes the assistant's text response, $X^{(*)}_{query}$ the input query under any modality configuration in Eq. 1, and $L$ the response length. Beginning with **Alignment Stage**, we use ASR as the alignment objective on LLaSO-Align, where each example contains a text instruction $X^t_{instruct}$, an audio input $X^a_{input}$, and its transcript $X^t_{answer}$, optimized with the objective in Eq. 2. During this stage we freeze the speech encoder and the LLM, updating only the projector $F_{proj}$ so that $H^a = F_{proj}(Z^a)$ aligns with the pre-trained LLM word embedding space, thereby establishing cross-modal semantic consistency for the next stage. In **Instruction Tuning Stage**, we then train on LLaSO-Instruct to endow the model with compositional instruction-following across linguistic, semantic, and paralinguistic tasks. The encoder remains frozen while we optimize $F_{proj}$ and $F_{llm}$ under Eq. 2, using the unified query formats in Eq. 1 and always producing textual responses. We provide training details in Appendix K.

| | Linguistic Task Category | | Semantic Task Category | | | | | |
| | **< Textual Instruction, Audio Input >** | **< Pure Audio >** | | **< Textual Instruction, Audio Input >** | | **< Audio Instruction, Textual Input >** | |
| **Modality Format:** | | | | | | | |
| **Tasks** | **ASR** | **AQA** | | | | | |
| Qwen2-Audio | 0.22 0.12 | 2.41 2.42 2.73 2.78 | 2.59 | 2.56 3.49 2.13 3.14 3.13 2.20 | 2.82 | 3.47 3.62 3.29 1.29 3.14 2.52 | 2.89 |
| Typhoon-Audio | 0.11 0.06 | 1.76 1.77 2.16 2.22 | 1.98 | 1.87 3.14 1.61 2.83 3.04 2.36 | 2.60 | 2.69 2.91 2.47 1.68 3.04 1.91 | 2.45 |
| Salmonn | 0.86 0.69 | 1.47 1.41 1.41 1.72 | 1.50 | 2.05 3.13 1.42 2.96 3.12 2.37 | 2.60 | 2.04 3.03 2.42 1.83 3.19 1.58 | 2.35 |
| Glm-4-Voice | 0.93 0.79 | 2.22 2.34 **3.29** 2.93 | 2.70 | 2.49 3.21 2.51 3.11 2.82 1.97 | 2.72 | 3.09 **4.06** 1.68 1.03 3.10 1.98 | 2.49 |
| Mini-Omni | 0.95 0.81 | 1.42 1.47 1.75 1.45 | 1.52 | 1.63 1.54 1.22 2.34 1.33 1.41 | 1.57 | 1.42 1.32 1.17 1.21 1.27 1.20 | 1.27 |
| Mini-Omni2 | 0.95 0.80 | 1.57 1.53 2.05 1.51 | 1.67 | 1.66 1.64 1.26 2.52 1.42 1.43 | 1.65 | 1.68 1.50 1.41 1.29 1.31 1.28 | 1.41 |
| Llama-Omni | 0.88 0.73 | 1.97 2.02 2.99 2.48 | 2.37 | 2.38 2.95 1.88 3.16 2.72 2.20 | 2.58 | 2.73 3.78 2.29 1.11 3.08 2.09 | 2.51 |
| Audio-Reasoner | 0.28 0.12 | 2.44 2.24 2.51 2.86 | 2.51 | 2.22 3.42 2.12 3.07 2.91 2.14 | 2.73 | 2.84 3.95 2.88 1.54 3.13 2.09 | 2.74 |
| Kimi-Audio | 0.14 0.05 | **2.94** 2.70 3.22 **3.45** | 3.08 | **3.28** 3.77 **3.35** **3.53** **3.38** 2.71 | 3.35 | **3.69** 4.01 3.38 1.16 3.16 2.77 | 3.03 |
| Qwen2.5-Omni | 0.40 0.26 | **2.94** **3.09** 3.22 2.63 | 2.97 | 2.99 **3.80** 3.20 2.96 3.19 2.12 | 3.05 | 3.46 3.88 **3.58** 1.19 3.15 2.42 | 2.95 |
| LLaSO-Base (Ours) | **0.08** **0.03** | 2.06 1.80 2.39 1.46 | 1.93 | 2.57 2.48 1.71 2.74 3.05 **2.90** | 2.58 | 2.72 2.62 2.28 **2.23** **3.74** 2.60 | 2.70 |
| **Metrics** | **WER↓** **CER↓** | **GPT-4o↑** | **Avg.GPT-4o↑** | **GPT-4o↑** | **Avg.GPT-4o↑** | **GPT-4o↑** | **Avg.GPT-4o↑** |

Table 2: Comparison of 11 LSLMs on LLaSO-Eval linguistic (ASR) and semantic (AQA) tasks across three modality configurations. ASR is evaluated by WER/CER (lower ↓ is better); AQA is scored by GPT-4o (higher ↑ is better). Cell shading ▨, ▨, and no shading denotes each model's relative ranking in a given modality (best → worst) by average GPT-4o score.

| | Paralinguistic Task Category — Speaker-centric — Modality Format: < Textual Instruction, Audio Input > | | | | | | | | | | | | Paralinguistic Task Category — Content-centric — Modality Format: < Textual Instruction, Audio Input > | | | | | | | |
| **Tasks** | **SGC** | **AC** | **AR** | **EIE** | **ER** | **SSD** | **SV** | **PSWL** | **PSSL** | **PR** | **SCR** | | **IP** | **EE** | **VSC** | **IC** | **ISC** | **PP** | **VC** |
| Qwen2-Audio | **1.00** 0.95 0.67 **0.99** 0.16 0.12 0.05 0.23 **0.52** | 18.69 | 0.54 0.31 | 0.24 0.30 | 0.43 0.25 | 1.86 | 1.95 0.17 | 1.19 | 2.60 | 2.40 | 3.04 2.52 2.73 0.85 **0.60** **0.60** | 19.02 | 0.02 |
| Typhoon-Audio | 0.85 0.77 0.59 0.67 0.21 0.14 0.11 0.10 0.12 | 20.47 | 0.40 0.24 | 0.28 0.12 | 0.46 0.20 | 2.04 | 1.71 0.33 | 3.08 | 0.98 | 0.85 | 3.13 1.86 2.85 0.49 0.16 0.16 | 36.83 | 0.17 |
| Salmonn | 0.59 0.44 0.13 0.18 0.22 0.32 0.10 0.26 0.06 | 11.24 | 0.31 0.24 | 0.30 0.21 | 0.50 0.19 | 1.32 | 1.38 0.13 | 1.82 | 1.09 | 0.75 | 4.07 1.88 3.29 0.61 0.16 0.16 | 41.92 | **0.22** |
| Glm-4-Voice | 0.11 0.12 0.04 0.07 0.07 0.09 0.03 0.02 0.01 | 15.35 | 0.13 0.08 | 0.14 0.02 | 0.10 0.04 | 1.62 | 1.84 0.24 | 0.90 | 1.00 | 0.98 | 1.85 1.78 2.34 0.32 0.00 0.03 | 40.20 | 0.08 |
| Mini-Omni | 0.14 0.00 0.00 0.00 0.00 0.06 0.00 0.02 0.01 | 21.34 | 0.04 0.04 | 0.04 0.07 | 0.11 0.03 | 1.24 | 1.46 0.00 | 0.92 | 1.00 | 0.98 | 1.39 1.26 1.42 0.02 0.01 0.06 | 61.49 | 0.00 |
| Mini-Omni2 | 0.11 0.00 0.02 0.00 0.00 0.03 0.00 0.00 0.00 | 18.46 | 0.03 0.06 | 0.00 0.01 | 0.12 0.03 | 1.16 | 1.54 0.00 | 0.97 | 1.00 | 0.97 | 1.89 1.26 1.60 0.03 0.01 0.03 | 59.32 | 0.00 |
| Llama-Omni | 0.36 0.26 0.03 0.26 0.07 0.14 0.07 0.16 0.17 | Reject | 0.31 0.08 | 0.16 0.04 | 0.26 0.08 | 1.28 | 1.38 0.13 | 1.46 | 20.19 | 21.11 | 1.58 1.80 2.34 0.03 0.00 0.12 | Reject | 0.07 |
| Audio-Reasoner | 0.38 0.32 0.38 0.37 0.14 0.18 0.02 0.23 0.12 | 13.57 | 0.52 0.29 | 0.32 0.28 | 0.35 0.16 | 2.61 | 2.03 0.09 | 1.08 | 2.40 | 1.84 | 4.10 2.29 **3.78** 0.59 0.20 0.17 | 32.68 | 0.15 |
| Kimi-Audio | 0.98 0.97 0.66 0.81 0.38 0.31 0.20 0.17 0.12 | 12.07 | **0.65** 0.34 | **0.52** 0.32 | 0.63 0.22 | **3.30** | 2.76 0.20 | 1.58 | 1.00 | 0.31 | 4.56 2.05 3.57 0.84 0.26 0.38 | 31.64 | 0.19 |
| Qwen2.5-Omni | 0.53 0.41 0.40 0.35 0.06 0.19 0.02 0.11 0.08 | **10.31** | 0.52 0.27 | 0.29 **0.33** | 0.42 0.15 | 1.25 | 2.12 0.27 | 1.28 | 3.53 | 3.52 | 3.91 2.00 2.78 **0.92** 0.51 0.44 | 18.37 | 0.12 |
| LLaSO-Base | 0.96 **0.99** **0.76** 0.91 **0.52** **0.83** **0.73** **0.70** 0.50 | 10.32 | 0.48 **0.48** | 0.17 0.26 | **0.99** **0.32** | 2.90 | **2.80** **0.39** | **0.03** | **0.04** | **0.02** | 4.86 **3.93** 3.57 0.78 0.50 **0.60** | **8.02** | 0.18 |
| **Metrics** | **ACC↑** | **MAE↓** | **ACC↑** | | | **GPT-4o↑** | **ACC↑** | **PER↓** | **WER↓** | **CER↓** | **GPT-4o↑** / **ACC↑** | **MAE↓** | **ACC↑** |

*%abstention*

Table 3: Performance of 11 LSLMs on LLaSO-Eval paralinguistic tasks, split by speaker-centric and content-centric groups. Cells are colored by abstention rate, as indicated by the color bar. Abstention rates were computed across all closed-ended tasks. Results are for the text instruction with audio input modality, reflecting the modality used for paralinguistic task training; our released datasets also include pure audio modality format samples for these tasks. *Reject* denotes 95% or more abstentions in a given task after manual inspection in open-ended settings/tasks.

# 5 EXPERIMENTS

## 5.1 SETUP

All experiments are conducted on LLaSO-Eval using the splits and task configurations defined in Section 3.4 and Appendix R. We benchmark LLaSO-Base against representative speech-language models, including Qwen2-Audio Chu et al. (2024), Typhoon-Audio Manakul et al. (2024), Salmonn Tang et al. (2024), GLM-4-Voice Zeng et al. (2024a), Mini-Omni Xie & Wu (2024b), Mini-Omni2 Xie & Wu (2024a), Llama-Omni Fang et al. (2025), Audio-Reasoner Xie et al. (2025), Kimi-Audio KimiTeam et al. (2025), and Qwen2.5-Omni Xu et al. (2025), running official checkpoints. Detailed versions and access methods for all baselines are provided in Appendix I.

## 5.2 METRICS

LLaSO-Eval spans a wide range of tasks, which we categorize into open-ended and closed-ended formats, defined as in Table 11. To support this diversity, we employ 7 evaluation metrics, with task-metric assignments summarized in Table 12. By default, metric scores are computed directly from the raw model outputs. For tasks where models generate free-form responses, we apply an answer-extraction step prior to evaluation. For open-ended tasks evaluion, we primarily use the GPT-4o Score, ranges from integer 1 to 5, providing a holistic assessment of response quality. In some cases, traditional metrics are also used to supplement GPT-4o. It is worth noting that certain tasks with well-established evaluation standards or clearly defined targets are better served by traditional metrics like PER or MAE. For closed-ended tasks, we use Accuracy and additionally track Abstention Rate for invalid or noncompliant answers. Complete metric details are provided in Appendix C.

## 5.3 RESULTS AND ANALYSIS

For cross-metric comparability, we report normalized overall and per-task performance in Figure 1 (Middle and Right), with full per-task scores and closed-ended abstention rates detailed in Table 2 and 3. LLaSO-Base attains the highest normalized overall score (0.72 vs. 0.65 for the next best model) and performs better on most individual tasks. The results establish the LLaSO Corpus as a reliable source for building reference models. LLaSO-Base, while not designed for state-of-the-art performance, exemplifies the power of an open, extensible, and reproducible workflow.

**Broader training coverage improves overall quality.** For deeper analysis we visualize Figure 6 comparing the performance and abstention rates of 11 evaluated models as a function of task coverage defined as the number of training tasks; for models with private, incomplete, or ambiguously reported training data, we use the number of evaluation tasks as a proxy. As can be seen, models trained on broader task sets consistently outperform task-focused systems in both overall and closed-ended performance, while also exhibiting fewer abstentions. This pattern highlights persistent difficulties on unseen tasks and underscores the importance of diversifying task coverage to improve performance and lower abstention rates.

**LSLMs may prefer content related tasks.** To further analyze model performance within paralinguistic tasks, we present a dumbbell plot in Figure 7, contrasting content-centric and speaker-centric results for each model. We observe that most models achieve higher performance and lower abstention rates on content-centric tasks than on speaker-centric ones. This disparity likely arises because content-centric tasks are more tightly linked to the semantic content, which LLM-based decoders are naturally equipped to process. In contrast, speaker-centric tasks demand more nuanced inference of latent speaker attributes, posing a greater challenge for current LSLMs and highlighting an important area for future improvement.

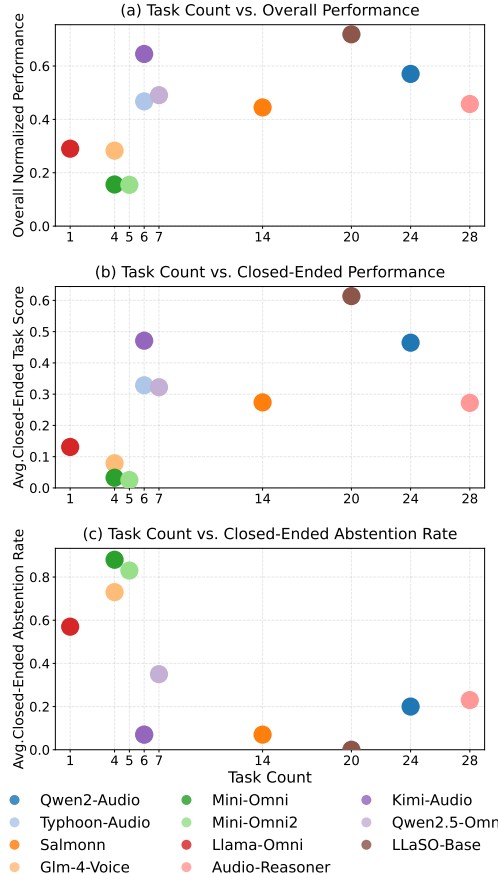

Figure 6: Task coverage vs. model performance and abstention. Each scatter plot shows 11 models by *Task Count*. *(a)* Overall performance (min–max normalized over all LLaSO-Eval tasks, cf. Figure 1 (Middle). *(b)* Average Closed-ended task performance. *(c)* Average abstention rate on closed-ended tasks. Closed-ended scores and abstention rates are calculate donly on tasks that require categorical selection. Higher scores indicate better performance; lower abstention rates indicate stronger instruction following performance.

**Generalization remains fragile, especially on unseen modalities.** Extending the observation from task coverage to modality, we find that most models underperform when evaluated across the three modality settings in Table 2. This weakness is unsurprising given that many baselines, as detailed in Appendix I, were trained to support only one or two input-output formats rather than the full spectrum. In particular, the <audio instruction, text input> configuration consistently lags behind the more common <text instruction, audio input>, even though the former should in principle be no harder for humans, as speech instructions are typically brief and the main content remains directly readable as text. A notable exception is Qwen2-Audio and its variant Audio-Reasoner, which achieve comparable results across the two formats.

**Pure audio remains the most challenging modality.** To better visualize the tendency, we present Figure 8 (Bottom). Even more striking, we observe that most models perform substantially *worse* on <pure audio> inputs than on <text instruction, audio input>, even for systems explicitly trained

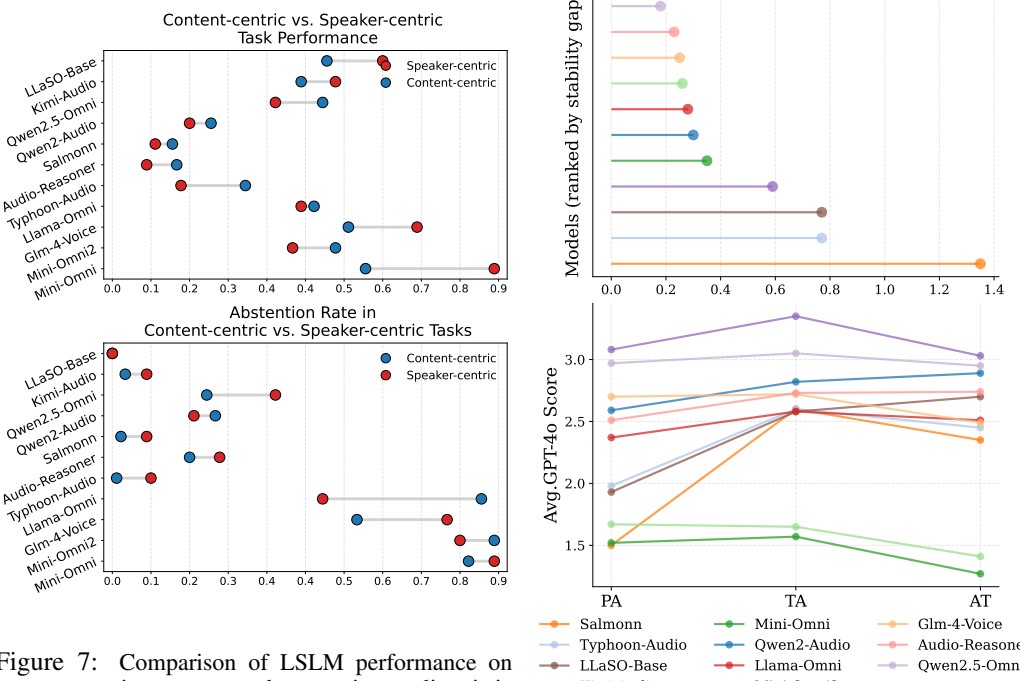

Figure 7: Comparison of LSLM performance on content-centric versus speaker-centric paralinguistic tasks. *Top:* For each model, min-max normalized performance scores are shown on content-centric (blue) and speaker-centric (red) tasks, with dumbbell lines indicating the magnitude and direction of intra-model performance differences. *Bottom:* Average abstention rates (lower is better) for closed-ended tasks in the same two centrics within the paralinguistic category. All evaluations are conducted under the text instruction paired with audio input configuration.

Figure 8: Stability and modality-wise performance of LSLMs. *Top:* Cross-modality stability, measured as the sum of absolute differences between a model's GPT-4o score on text + audio (TA) and the other two formats, pure audio (PA) and audio + text (AT); lower values indicate greater robustness. *Bottom:* Average GPT-4o scores across the three configurations. Colors are consistent across plots.

on speech-to-speech or spoken-query QA, with drops sometimes exceeding those on unseen configurations. Interestingly, a few models such as Qwen2.5-Omni, GLM-4-Voice, and the Mini-Omni family achieve comparable performance across modalities. To quantify this, we measure cross-modality stability as the sum of absolute performance differences between the common text with audio setting and the other two formats, i.e., $|TA - PA| + |TA - AT|$, and report models in ascending order of stability in Figure 8 (Top). We find that **interleaving** and **parallel decoding** substantially reduce modality gaps, where the top 8 among 11 systems, excluding Qwen2-Audio and its variant, adopt these strategies and exhibit notably smaller disparities. Although the outliers likely reflect factors beyond modality combination design, These results highlight interleaving and parallel decoding as promising directions for improving cross-modal generalization. Further modality- and task-level analyses, case studies across different task and modality, and model-specific discussions are provided in Appendix D, F, and G.

## 6 CONCLUSION

Despite recent advances, progress in LSLMs has been constrained by fragmented resources, limited task diversity, and a lack of standardized evaluation. To address these challenges, we present LLaSO: the first fully open, end-to-end framework for LSLM development. It comprises 25.5M samples for alignment and instruction tuning (LLaSO-Align and LLaSO-Instruct), a stratified benchmark of 15K samples (LLaSO-Eval), and a reproducible 3.8B-parameter reference model (LLaSO-Base) successfully verified proven vision-language architecture to speech domain. Our evaluation further reveals that, although broader task coverage improves overall performance, current LSLMs still face notable generalization gaps on unseen tasks and pure-audio settings. Encouragingly, models employing interleaving or parallel decoding demonstrate improved robustness in these challenging scenarios, highlighting promising directions for future research. By releasing all data, benchmarks, and models, LLaSO lowers the barrier to entry and provides a foundation for systematic, community-driven progress in large-scale speech-language modeling. *See Supplementary Material for reproducibility.*

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

## A  LIMITATION

While our work establishes a unified, open-source foundation for compositional speech-language instruction tuning, several limitations remain. First, LLaSO Corpus is currently limited to English, which constrains its direct applicability to non-English and low-resource languages. Extending the dataset and benchmark to multilingual scenarios is an important direction for achieving broader impact and inclusivity. Second, despite surpassing prior datasets in task diversity and modality coverage, the granularity and availability of source materials inherently influence our corpus composition, particularly in the underrepresented paralinguistic categories and rare interaction scenarios. Third, our reference model, LLaSO-Base, intentionally prioritizes reproducibility and extensibility over achieving SOTA performance. Consequently, its architecture and model size (3.8 billion parameters) are modest compared to larger models, and our evaluations have primarily included similarly sized or smaller baselines. Assessing and benchmarking significantly larger LSLMs would provide further insights into scaling behaviors and capabilities. Fourth, certain challenging multimodal interactions such as open-ended dialogues involving overlapping speech, or zero-shot generalization to entirely new domains are only partially addressed within our current benchmark and model architecture. We encourage the research community to build upon our foundation to tackle these limitations, further refining instruction-tuned speech-language models for diverse languages, scenarios, and real-world applications.

## B  ETHICAL STATEMENT

### B.1  DATA PRIVACY AND CONSENT

All training and evaluation data are sourced solely from publicly available datasets, with no use of private or personally identifiable information. Synthetic, TTS, and sound effect samples contain no human-identifiable content. No re-identification or de-anonymization was performed at any stage. All data handling complies with ethical standards and legal requirements.

### B.2  LICENSING AND RESPONSIBLE USE

All data, code, and model weights are released under permissive open-source licenses, with explicit terms governing use and redistribution. The resources are intended for academic, non-commercial research, and must be used in accordance with ethical standards and applicable copyright laws.

### B.3  DIVERSITY AND REPRESENTATIVENESS

We strive for diversity in gender, age, accent, language, and emotion across both collected and synthetic data, employing balanced sampling where possible. Nonetheless, certain groups and languages remain underrepresented, and we acknowledge the risk of bias. We encourage the community to further augment and improve coverage. All datasets and models are released without representing the views or interests of any particular group or institution.

### B.4  FAIRNESS AND MISUSE PREVENTION

Our models and datasets may exhibit uneven performance across different tasks, languages, or demographic groups, and should not be considered universally fair or unbiased. We explicitly prohibit the use of our work for surveillance, discrimination, harassment, or any activities that may harm individuals or communities. We encourage responsible research and deployment that respects the rights and dignity of all users.

# C  METRIC DETAILS

Given the diversity of tasks and modalities in LLaSO-Eval, we define 7 metrics to ensure comprehensive evaluation. During evaluation, by default metric scores are computed directly from the complete model outputs. However, certain tasks require an intermediate step of extracting structured answers from the model outputs prior to evaluation; such cases are explicitly noted in their respective metric descriptions below. Task-specific metric assignments are detailed in Table 12.

**WER and CER.**  Word Error Rate (WER) and Character Error Rate (CER) Morris et al. (2004); huggingface (2023); Chen et al. (1998) quantify transcription accuracy derived from Levenshtein distance Navarro (2001) between the model prediction and the ground-truth transcript. WER operates at the word level, while CER operates at the character level. Both metrics are employed for ASR and SCR tasks. Lower values indicate better accuracy. Typically, WER and CER scores range from 0 (perfect match) to 1, although values exceeding 1 can occur due to excessive insertions or substitutions.

**PER.**  Phoneme Error Rate (PER) is analogous to WER and CER but specifically measures the Levenshtein distance between the predicted and ground-truth phoneme sequences with brianlan (2017), providing a phoneme-level accuracy assessment. Similar to WER and CER, lower PER values indicate superior performance, typically ranging from 0 upwards, with 0 representing a perfect phoneme prediction. We apply this metric exclusively to the PR task.

**Accuracy.**  Accuracy is defined as the proportion of exact matches between the model's prediction and the ground-truth label. This metric is applied to all closed-ended tasks, as specified in Table 11, which also indicates which tasks are open- versus closed-ended. For closed-ended tasks, the model must select a single answer from a predefined label set, and a response is marked correct only if it precisely matches the reference label; predictions containing multiple candidate labels or irrelevant content are treated as incorrect. Accuracy ranges from 0 to 1, with higher values indicating better performance. Additionally, the metric is also computed for the open-ended PSSL task, where models rate sentence-level pronunciation across three dimensions, accuracy, prosodic, and fluency. Given that model outputs are typically free-form, we use regular expressions to extract numeric scores from responses, accommodating variations such as "accuracy is 8", "fluency: 7", or "9 for prosodic". Responses providing valid numeric scores for all three dimensions are retained; others are excluded. For each sample, we compute the average of the exact-match accuracies across these three dimensions, then report the overall accuracy averaged across all evaluated samples. Further, we complement this rule-based measure with an additional GPT-4o evaluation to ensure comprehensive assessment.

**MAE.**  Mean Absolute Error (MAE) is adopted for tasks requiring numerical predictions, such as AR and PR. In these tasks, the model is explicitly instructed to generate a single numeric value. However, many LSLMs produce free-form textual outputs rather than direct numeric predictions Adlakha et al. (2024), necessitating an answer extraction procedure prior to metric calculation. For the AR task, the predicted numeric value represents age and thus must be extracted reliably from the model output. Employing a regular expression, our script initially attempts a direct integer conversion; if unsuccessful, it searches for numeric patterns, and if a numeric range like "40-45" is detected, it computes the rounded average of the two endpoints. For outputs containing descriptive keywords "adult" without numeric information, we substitute a canonical age value 22. The PR task requires evaluation of the model's ability to predict MIDI note values ranging from 0 to 127. Specifically, our extraction function sequentially attempts integer conversion, rounded float conversion, and finally, averaging numeric ranges. Strings containing pitch-related keywords (e.g., "midi", "pitch", "hz") but lacking numeric values are marked as invalid predictions. Only predictions within the MIDI range of 0 to 127 are considered valid for metric computation. In all cases, if a valid numeric value cannot be extracted from either the model's prediction, that instance is omitted from the calculation. The final MAE is computed as the mean absolute difference between extracted predictions and ground-truth numeric values across all valid instances. Lower MAE values indicate better numerical prediction performance.

**GPT-4o Score.** For AQA and other open-ended generative tasks, where model responses are unconstrained and may vary widely in form and content, thus we employ GPT-4o (OpenAI, gpt-4o-mini, Version 2024-07-18) as an automatic evaluator. Following a standardized evaluation template, GPT-4o assigns an integer score from 1 to 5, reflecting both the relevance and accuracy of the model's response relative to the reference answer. Further details of the evaluation prompt are provided in Appendix O, and task-specific metric assignments are summarized in Table 12.

**Abstention Rate.** Some LSLMs may abstain from answering tasks involving unfamiliar modality formats or instructions, fail to follow instructions, or explicitly state their inability to process audio. To quantify such behavior, we report the abstention rate for closed-ended tasks, defined as the proportion of responses in which the model either refuses to answer, returns irrelevant content, or fails to select a valid label from the predefined set. Higher scores indicate better performance; lower abstention rates indicate stronger instruction following. An abstention is counted whenever the model's output does not comply with the task requirement to select a label. Abstention rate is not reported for open-ended tasks, as their free-form nature precludes a rule-based criterion for abstention.

# D DETAILED ANALYSIS

We find that LSLMs perform poorly on unseen tasks and unfamiliar modality formats, with especially weak instruction-following in unseen tasks.

*Broader task coverage leads to better performance and lower abstention rates.* We present the overall performance, closed-ended task performance, and closed-ended task abstention rates for 11 models, alongside the number of tasks each model was trained on, in Figure 6. For models with private, incomplete, or ambiguously reported training data, we use the number of evaluation tasks as a proxy for task coverage. The results show that models exposed to a wider range of tasks achieve higher performance in both overall and closed-ended tasks, and fewer abstentions. This finding suggests that, in creating LSLMs for speech understanding, one should diversify the tasks as much as possible, to improve the model performance and reduce the abstention rate.

*LSLMs perform worse on unseen modality configuration.* Most existing LSLMs only support one or two modality configurations. We evaluate their generalization across three different input formats. Specifically, we select representative models and calculate their average performance on AQA task across all major input modality configurations with results summarized in Table 2. We observe that model performance in the audio instruction with text input setting drops consistently compared to the familiar text instruction with audio input configuration. At the same time we find that Qwen2-Audio is an outlier, showing that it and its variant Audio-Reasoner obtain similar results in both formats. Notably, from a human perspective, audio instruction with text input should be no more difficult and is arguably even simpler, since only the (typically brief) instruction needs to be heard, while the main input remains directly readable as text, as illustrated in Figure 4 (b, c). Nonetheless, our findings demonstrate that LSLMs still struggle with modality configurations outside their explicit training coverage.

*Pure audio modality configuration may still challenging.* We illustrate the performance of 11 models three major modality formats in Figure 8 (Bottom). In most cases, models demonstrate substantially *lower* performance on pure audio formats than on the more common text instruction with audio input setting, even when they are explicitly trained to handle pure audio via speech-to-speech or spoken-query-based QA (SQQA) tasks. Notably, for some of these models, the performance drop from text with audio to pure audio is even greater than the decline observed on modality formats they have never seen during training, such as audio instruction with text input. Interestingly, only a handful of models such as Qwen2.5-Omni, GLM-4-Voice, and the Mini-Omni family achieve comparable performance across pure audio and text + audio modalities. Nonetheless, for most current LSLMs, the pure audio configuration remains a notably challenging setting.

*Interleaving and parallel decoding strategies help bridge performance gaps across modality configurations.* As shown in Table 2 and Figure 8 (Bottom), nearly all models achieve their best results on the common text instruction with audio input setting. To assess model robustness to modality shifts, we compute the sum of absolute performance differences between this common configuration and the other two input formats. We present the results in ascending order of stability in Figure 8 (Top). Among the eleven models evaluated, the top eight with the exception of Qwen2-Audio and its variant employ interleaving or parallel decoding strategies (see Appendix I for benchmarking model details), and exhibit notably reduced modality gaps. These outliers may reflect factors outside of modality combination design. Overall, our results provide empirical evidence that interleaving and parallel decoding can bridge the performance gap between text and audio modalities.

*LSLMs may prefer content related tasks.* To further analyze model performance on paralinguistic tasks, we present a dumbbell plot in Figure 7, contrasting content-centric and speaker-centric results for each model. We observe that most models achieve higher performance and lower abstention rates on content-centric tasks than on speaker-centric ones. This disparity likely arises because content-centric tasks are more tightly linked to the semantic content, which LLM-based decoders are naturally equipped to process. In contrast, speaker-centric tasks demand more nuanced inference of latent speaker attributes, posing a greater challenge for current LSLMs and highlighting an important area for future improvement.

# E ABLATION

We conduct ablation experiments on different training strategies in LLaSO Corpus, as shown in Table 4 and 5. (i) *Alignment Robustness.* We evaluate ASR performance both immediately after the alignment stage and following the subsequent instruction-tuning phase. After alignment, the model achieves strong results (WER = 0.05, CER = 0.01). After multi-task instruction tuning, ASR performance declines slightly (WER = 0.08, CER = 0.03), yet remains competitive, due to we include ASR samples within the LLaSO-Instruct dataset for mitigating catastrophic forgetting. (ii) *Encoder Fine-tuning.* We ablate the effect of unfreezing the audio encoder during the instruction-following stage, comparing the results of freezing versus jointly training the encoder, projector, and LLM on LLaSO-Instruct. When the encoder is unfrozen, ASR performance drops more substantially (WER = 0.14, CER = 0.07), relative to the frozen configuration. In contrast, AQA (semantic) tasks see modest improvements (see Table 5), while paralinguistic tasks exhibit a slight decline (see Table 4). This suggests that while joint fine-tuning may benefit certain high-level reasoning tasks, it may compromise low-level speech recognition and nuanced paralinguistic abilities.

| *Paralinguistic Task Category* | | | | | | | | | |
|---|---|---|---|---|---|---|---|---|---|
| **Speaker-Centric** | | | | | **Content-Centric** | | | | |
| < Text, Audio > | | | | | | | | | |
| **Tasks** | **LLaSO-Base** | **LLaSO-Base (Unfrozen)** | **Δ (U-F)** | **Metrics** | **Tasks** | **LLaSO-Base** | **LLaSO-Base (Unfrozen)** | **Δ (U-F)** | **Metrics** |
| | 0.96 | 0.88 | -0.08↓ | | **PR** | 0.03 | 0.03 | 0.00= | **PER↓** |
| | 0.99 | 0.99 | 0.00= | **ACC↑** | | 0.04 | 0.05 | 0.01↓ | **WER↓** |
| **SGC** | 0.76 | 0.61 | -0.15↓ | | **SCR** | 0.02 | 0.04 | 0.02↓ | **CER↓** |
| | 0.91 | 0.97 | 0.06↑ | | | 4.86 | 4.80 | -0.06↓ | |
| | 0.91 | 0.86 | -0.05↓ | **Avg.ACC↑** | **IP** | 3.93 | 3.90 | -0.03↓ | **GPT-4o↑** |
| | 0.52 | 0.40 | -0.12↓ | | **EE** | 3.57 | 3.44 | -0.13↓ | |
| **AC** | 0.83 | 0.86 | 0.03↑ | **ACC↑** | **VSC** | 0.78 | 0.82 | 0.04↑ | |
| | 0.73 | 0.78 | 0.05↑ | | **IC** | 0.50 | 0.55 | 0.05↑ | **ACC↑** |
| | 0.69 | 0.68 | -0.01↓ | **Avg.ACC↑** | **ISC** | 0.60 | 0.77 | 0.17↑ | |
| | 0.70 | 0.68 | -0.02↓ | **ACC↑** | **PP** | 8.02 | 10.55 | 2.53↓ | **MAE↓** |
| **AR** | 0.50 | 0.38 | -0.12↓ | | **VC** | 0.18 | 0.20 | 0.02↑ | **ACC↑** |
| | 0.60 | 0.53 | -0.07↓ | **Avg.ACC↑** | *Linguistic Task Category* | | | | |
| | 10.32 | 8.78 | -1.54↑ | **MAE↓** | < Text, Audio > | | | | |
| | 0.48 | 0.45 | -0.03↓ | **ACC↑** | **Tasks** | **LLaSO-Base** | **LLaSO-Base (Unfrozen)** | **Δ (U-F)** | **Metrics** |
| **EIE** | 0.48 | 0.37 | -0.11↓ | | | 0.08 | 0.14 | 0.06↓ | **WER↓** |
| | 0.48 | 0.41 | -0.07↓ | **Avg.ACC↑** | **ASR** | 0.03 | 0.07 | 0.04↓ | **CER↓** |
| | 0.17 | 0.16 | -0.01↓ | **ACC↑** | **Tasks** | **LLaSO-Base (Aligned)** | **LLaSO-Base** | **Δ (F-A)** | **Metrics** |
| **ER** | 0.26 | 0.30 | 0.04↑ | | | 0.05 | 0.08 | 0.03↓ | **WER↓** |
| | 0.22 | 0.23 | 0.01↑ | **Avg.ACC↑** | **ASR** | 0.01 | 0.03 | 0.02↓ | **CER↓** |
| **SSD** | 0.99 | 0.99 | 0.00= | **ACC↑** | | | | | |
| **SV** | 0.32 | 0.16 | -0.16↓ | **ACC↑** | | | | | |
| **PSWL** | 2.90 | 2.68 | -0.22↓ | **GPT-4o↑** | | | | | |
| **PSSL** | 0.39 | 0.24 | -0.15↓ | **ACC↑** | | | | | |
| | 2.80 | 2.66 | -0.14↓ | **GPT-4o↑** | | | | | |

Table 4: Ablation results for LLaSO-Base on paralinguistic (speaker-centric and content-centric) and linguistic tasks, all evaluated under the text instruction with audio input modality. Frozen (F) and unfrozen (U) refer to whether the audio encoder is fixed or updated during instruction tuning, respectively. Δ (U-F) reports the performance change between unfrozen and frozen encoder variants during finetuning, while Δ (F-A) compares results after finetuning with frozen encoder (F) and after the alignment stage (A) for ASR. ↑/↓ denote gains or drops; metrics follow previous tables.

| Tasks | LLaSO-Base | LLaSO-Base (Unfrozen) | Δ (U-F) | Metrics |
|---|---|---|---|---|
| | *Semantic Task Category* | | | |
| | < Pure Audio > | | | |
| AQA | 2.06 | 2.27 | 0.21↑ | |
| | 1.80 | 2.27 | 0.47↑ | |
| | 2.39 | 2.23 | -0.16↓ | **GPT-4o**↑ |
| | 1.46 | 1.98 | 0.52↑ | |
| | 1.93 | 2.19 | 0.26↑ | **Avg.GPT-4o**↑ |
| | < Text, Audio > | | | |
| AQA | 2.57 | 2.54 | -0.03↓ | |
| | 2.48 | 2.42 | -0.06↓ | |
| | 1.71 | 1.96 | 0.25↑ | |
| | 2.74 | 2.80 | 0.06↑ | **GPT-4o**↑ |
| | 3.05 | 3.09 | 0.04↑ | |
| | 2.90 | 2.87 | -0.03↓ | |
| | 2.58 | 2.61 | 0.03↑ | **Avg.GPT-4o**↑ |
| | < Audio, Text > | | | |
| AQA | 2.72 | 3.22 | 0.50↑ | |
| | 2.62 | 2.84 | 0.22↑ | |
| | 2.28 | 2.47 | 0.19↑ | |
| | 2.23 | 2.43 | 0.20↑ | **GPT-4o**↑ |
| | 3.74 | 3.68 | -0.06↓ | |
| | 2.60 | 3.26 | 0.66↑ | |
| | 2.70 | 2.98 | 0.28↑ | **Avg.GPT-4o**↑ |

Table 5: Ablation results for LLaSO-Base, comparing frozen (F) and unfrozen (U) audio encoder variants during instruction tuning. The table reports Δ (U-F) performance changes on AQA tasks across three major modality configurations; ↑ / ↓ denote relative gains or drops. Metrics follow earlier tables.

# F  CASE STUDY

We provide qualitative evidence of the compositional flexibility and unified modeling offered by our framework. Figure 4 demonstrates that LLaSO-Base seamlessly accommodates all three instruction-input modality pairings. In particular, the pure audio example highlights the system's ability to disentangle instructions from content solely within the audio stream. For the other formats, LLaSO-Base reliably grounds reasoning and response generation in the correct modality, adapting to instructions and content presented in any combination. We present some task prototypes unified by our system in Figure 3 and present more cases across different instruction-input structures and tasks in Appendix J. These examples show that, unlike models limited rigid instruction-input structures, LLaSO-Base generalizes across both categorical and compositional tasks without requiring task-specific modules or post-processing.

To better understand the challenges across modality configurations and tasks, we further present cases under the same benchmark. As shown in Figure 9 (I), we sample three representative cases for Salmonn across the primary modality configurations. In case (I)(b) with textual instruction and audio input, the model's familiar modality configuration, Salmonn correctly follows the user instruction. The textual instruction asks the model to determine whether the first sentence in the speech content can be used to define the term in the second sentence, and to answer "yes" or "no". The audio provides a definition of "database" in the first sentence and describes "MySQL" in the second. The model correctly interprets both the instruction and the speech, and returns the correct answer "yes". In contrast, when the modality configuration shifts outside the model's primary training distribution, distinct failures emerge. Under the pure audio setting (I)(a) the model receives both the instruction and content as audio, yet responds with a counter-question: "What is the answer to the

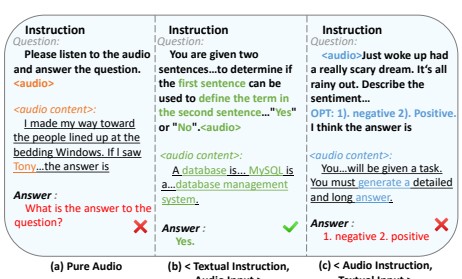

(I) Three evaluated cases from Salmonn across the primary modality configurations. The correct one corresponds to the model's supported training format; the two errors are from modality formats with limited or no training exposure.

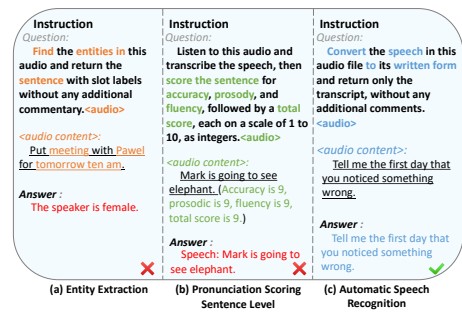

(III) Three samples from Typhoon-Audio: one from a well-represented task with correct prediction, and two errors from tasks absent in its training.

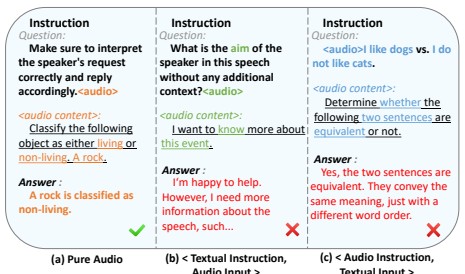

(II) Three Llama-Omni responses sampled from the three major modality configurations. The correct example comes from its supported format; the two errors are from its unsupported modality configurations.

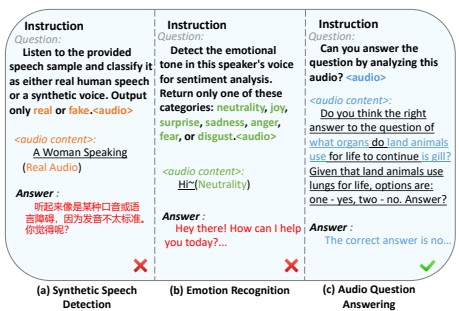

(IV) Selected answers from GLM-4-Voice illustrate success on a task with ample training exposure and failure on two tasks that fall outside its main training coverage.

Figure 9: Case studies from LLaSO-Eval. (Left) Model behaviors under different modality configurations (Salmonn and Llama-Omni), highlighting the importance of supporting multiple modality formats for reliable instruction following. (Right) Model behaviors across tasks with different coverage (Typhoon-Audio and GLM-4-Voice), underscoring the necessity of broad task coverage for generalization.

question?" This indicates that while the model has some prior exposure to SQQA tasks, it fails to correctly interpret or respond to this particular modality configuration where both instruction and content are delivered as audio. In Figure 9 (I)(c), where an audio instruction is followed by a textual input, the spoken instruction assigns a generic task, prompting the model to generate an answer, while the accompanying text presents a tweet and asks for its sentiment, providing two answer options. The relevant information for reasoning is contained within the text input, and the instruction directs the model to perform a classification. However, the model does not follow the instruction; instead, it merely repeats the sentiment options, "1. negative 2. positive", without making a decision. We observe similar results in Llama-Omni across modality formats, illustrated in Figure 9 (II). This model is primarily trained on pure audio modality, and this is directly reflected in the sampled cases. In (II)(a) where both the instruction and input are delivered as audio, the model answers successfully classifying the object as non-living, demonstrating effective handling of its core modality. Nonetheless, when presented with configurations outside this primary distribution, the model fails to execute the intended tasks. In the text plus audio modality format (II)(b), it is unable to infer the speaker's aim from the speech and instead requests further contextual details. Under (II)(c) the modality configuration of audio instruction paired with textual input, the model follows speech instruction but overlooks the explicit negation in the text input and incorrectly judges the two sentences as equivalent. Taken together, this observation highlights the importance of comprehensive modality coverage for multimodal instruction following.

To provide an intuitive comparison of model performance on covered versus uncovered tasks, we sample more cases for representative baselines in Figure 9. Figure 9 (III) illustrates this contrast for Typhoon-Audio model. In (III)(a) we present a sample from the entity extraction task, which

is not included in the model's training. Here, the query requests identification of entities from the speech, but the model misinterprets the task as speaker gender classification, responding with "The speaker is female." We present a pronunciation scoring sentence-level (PSSL) task sample in (III)(b), where the model is instructed to evaluate the speech for accuracy, prosody, and fluency. However, it only provides a plain transcription and omits the required scoring. In contrast, when evaluated on a task present in its training, the model demonstrates accurate performance. For the ASR task presented at (III)(c) the model successfully transcribes the speech to text as instructed without additional information. Similar results are evident with GLM-4-Voice in Figure 9 (IV). In (IV)(a) tasked with synthetic speech detection, the model avoids providing a categorical decision and instead produces an off-topic statement. When we prompt the model for emotion recognition, it generates a generic conversational reply, neglecting to engage with the specified sentiment classification task as in (IV)(b). Nevertheless, we can observe that the model successfully completes an audio question answering task in (IV)(c), owing to the presence of this task in its training. These findings underscore the essential role of comprehensive task coverage in building models across diverse speech-language tasks.

## G  DISCUSSION

In addition to our quantitative analysis, a closer manual inspection of model outputs reveals several distinct patterns and recurring issues that warrant discussion. For example, Qwen2-Audio occasionally misinterprets the SV task as SGC. On other tasks, Although this model achieves relatively strong quantitative scores on the open-ended PP task, qualitative check reveals that a large proportion of its outputs are empty strings (41 cases) or single periods (30 cases) among 112 samples in the text instruction plus audio input configuration. For the VC task, the abstention rate is especially high. In our manual review, 82 out of 100 samples in the same modality setting resulted in a single period ("."). As to Typhoon-Audio, it occasionally responds in Thai to English prompts, which is likely attributable to the inclusion of Thai data during fine-tuning. Salmonn, when presented with pure audio or audio plus text modality input, often refuses to answer, asks clarifying questions, or claims the audio contains no content; this may stem from its limited exposure to pure audio instruction data (approximately 20K SQQA task samples from WikiQA Yang et al. (2015)) and to the modality configuration of audio instruction with text input, restricting its ability to generalize. Additionally, in the ASR task, Salmonn's outputs for most samples consist entirely of uppercase English letters, which explains its poor quantitative performance on this task. GLM-4-Voice frequently generates responses in Chinese and at times misinterprets the audio input as part of the conversational context, rather than as content to be analyzed. A similar pattern is observed in the Mini-Omni family, which occasionally interprets the audio input, such as a speaker's utterance to be classified, as either an instruction or as primary content. For example, in ASR task with the text plus audio input configuration, approximately 43% of Mini-Omni and 53% of Mini-Omni2 responses begin with phrases like "It sounds like...", reflecting a tendency to treat the input as dialogue. Meanwhile Llama-Omni exhibits a high rate of refusals in tasks beyond AQA, suggesting limited coverage of tasks. Manual inspection of its ASR task outputs further reveals that over 10% of samples are explicit abstains, likely because the model was not trained on ASR data. These phenomena further illustrate the necessity for broad and balanced coverage across both tasks and modality configurations in model development. Audio-Reasoner presents a different set of challenges, often exhibiting hallucinated completions such as appending "the answer is A" or fabricating multiple-choice options like "E," "F," or "S." Since it is trained on Qwen2-Audio-Instruct with chain-of-thought data, this tendency may stem from exposure to reasoning-style supervision. Kimi-Audio, when performing the PR task, sometimes treats the input as an ASR query or outputs a sequence of isolated phonemes, which leads to lower evaluation scores. Qwen2.5-Omni occasionally shows similar confusion between PR and ASR, and routinely appends conversational phrases like "feel free to ask me more." While such additions might be intended to improve user interaction, they can undermine instruction following if overfitting, as some tasks in our benchmark explicitly requires models to return only the answer. Taken together, these observations offer practical insights into persistent issues in instruction following, task differentiation, and output format consistency. We hope these manual inspections are helpful for the community and can inform future model development and evaluation.

## H VOCAL MIXING STRATEGY

To generate multimodal variants from text-based semantic QA data, we employ advanced audio synthesis tools such as MeloTTS (Zhao et al., 2023), ChatTTS (2noise, 2024), and OpenVoice (Qin et al., 2023). These systems support controllable attributes (e.g., gender, speed, tone, accent), enabling the creation of audio segments with varied vocal styles. The synthesized audio is then combined with either text or additional audio segments to form the three modality configurations described in the main paper.

For example, in a pure-audio sample shown as Figure 10, one speaker may voice the instruction in a cheerful American accent, while another renders the task content in a neutral British tone, simulating realistic multi-speaker interaction. Similarly, in <Text Instruction, Audio Input> or <Audio Instruction, Text Input> formats, synthesized speech segments are paired with textual components to enrich acoustic diversity. Using distinct voices for instructions and content helps clearly separate roles within the dialogue and better approximates real-world speech-language interactions.

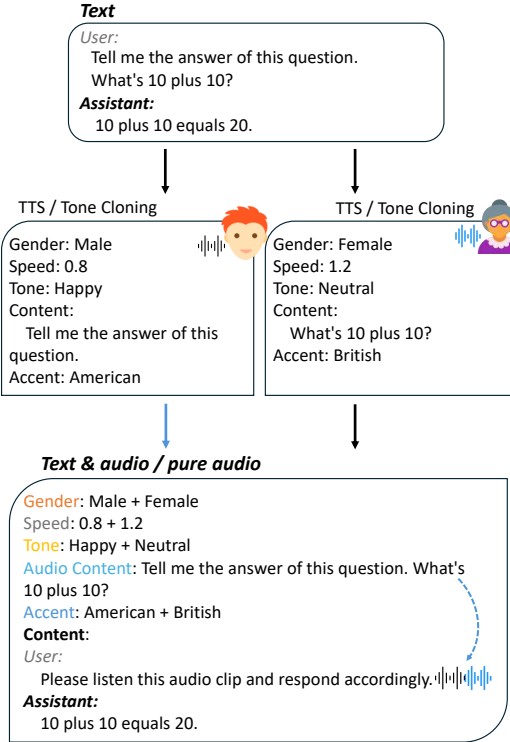

Figure 10: **Illustration of vocal style mixing.** Utterances are synthesized with varied speaker traits and applied across all three modality configurations, expanding acoustic diversity and simulating realistic multi-speaker scenarios.

## I BENCHMARKING CANDIDATES

**Qwen2-Audio.** A LSLM from Qwen Team designed for both audio analysis and voice chat. It integrates a Whisper-large-v3 audio encoder with a Qwen-7B language model, enabling processing of audio and text inputs for instruction following and conversational tasks. The model supports both audio + text and pure audio modality configurations, automatically distinguishing between analysis and dialogue modes without explicit prompts It achieves state-of-the-art results such as AIR-Bench and CoVoST2, with open-source demos, weights, and inference code.

**Typhoon-Audio.** A LSLM from SCB 10X and the University of Cambridge supporting both English and Thai. It integrates Whisper-large-v3 (fine-tuned for Thai) and BEATs audio encoders, a Q-

Former adapter, and a Typhoon-1.5-8B-Instruct LLM. The model supports both text-audio and pure audio (namely speech instruction following in this paper) configurations. Demo, model weights, and inference code are open-source.

**Salmonn.**   Salmonn is an unified LSLM with a dual-encoder architecture, Whisper and BEATs, linked via a window-level Query Transformer to a Vicuna-based LLM. The model is trained in three task levels using a two-stage alignment and instruction-tuning scheme, and further enhanced through activation tuning to unlock emergent capabilities. Salmonn supports audio-plus-text and pure audio (through SQQA) modality configurations and diverse task types. Demos, model checkpoints, training/inference code, and training data are all publicly available.

**Glm-4-Voice.**   An end-to-end spoken chatbot supporting both Chinese and English from Zhipu.AI and Tsinghua University. The model combines Glm-4-9B-Base with a supervised speech tokenizer and a flow-matching speech decoder, pre-trained on 1T tokens of speech-text and speech-only data. Fine-tuned with a streaming-thoughts template, it alternates between text and speech tokens for seamless, low-latency conversational output. GLM-4-Voice accepts speech or text inputs and produces simultaneous speech and text responses. Model weights, demo, and inference code are open-source.

**Mini-Omni.**   Developed by Inspirai and Tsinghua University, Mini-Omni is a streaming speech-to-speech conversational LLM integrating a Whisper-small encoder, modality adapters, a Qwen2-0.5B transformer language model, and a TTS adapter. The system employs parallel decoding for efficient, real-time, end-to-end speech input and streaming audio output. Model weights, inference code, demo, and the VoiceAssistant-400K dataset are open-source.

**Mini-Omni2.**   An omni-interactive multimodal model, developed by Inspirai and Tsinghua University as an upgraded version of Mini-Omni, combining CLIP (ViT-B/32) for vision, Whisper-small for audio, and Qwen2-0.5B for language. It enables real-time, end-to-end voice conversations with users, supporting image, audio, and text inputs and text, audio outputs. The model, inference and demo code are open-source.

**Llama-Omni.**   Developed by ICTNLPLab at CAS, this model integrates a frozen Whisper-large-v3 encoder, a trainable speech adaptor, a Llama-3.1-8B-Instruct language model, and a streaming speech decoder. Its key innovation is simultaneous generation of both text and speech responses from spoken instructions, enabling low-latency, end-to-end speech-to-text and speech-to-speech interaction. The model, along with its training data, weights, demo, and inference code, is open-source.

**Audio-Reasoner.**   A reasoning-oriented LSLM developed by fine-tuning Qwen2-Audio with structured chain-of-thought (CoT) supervision on its 1.2M-sample CoTA dataset. Emphasizing complex audio reasoning, it demonstrates the benefits of CoT-style instruction tuning, achieving competitive results including MMAU-mini and AIR-Bench-Chat. It is open-source along with its model checkpoint, demo, inference code, and dataset.

**Kimi-Audio.**   An audio foundation model developed by the Kimi Team featuring a hybrid architecture with an audio tokenizer, audio encoder, core audio LLM, parallel heads for both text and audio generation, and an audio detokenizer, using continuous acoustic vectors and discrete semantic tokens. Pre-trained on 13 million hours of diverse open and in-house audio, the model is fine-tuned for multimodal comprehension and generation tasks involving speech, music, and sound effects, including audio understanding, speech conversation, and audio-to-text chat. It demonstrates strong performance on benchmarks such as VoiceBench, VocalSound, and MELD. The project is open-source, providing demo data, fine-tuning and inference code, released model weights, and an audio evaluation toolkit.

**Qwen2.5-Omni.**   A unified end-to-end real-time multimodal model developed by the Qwen team, supporting text, audio, image, and video inputs, with both streaming text and speech outputs. Built on the Thinker-Talker architecture, it enables flexible cross-modal interactions and streaming, facilitated by TMRoPE and block-wise encoding for efficient temporal alignment. The model achieves

strong performance on diverse multimodal benchmarks like VoiceBench and MMAU and open-source with released weights, APIs, and inference code.

| Model Name in this Paper | Official Model Name | URL | #Params | Supported Modalities | Interleaving or Parallel Decoding |
|---|---|---|---|---|---|
| Qwen2-Audio | Qwen/Qwen2-Audio-7B-Instruct | [Model Card] | 7B | $\langle T, A \rangle$, $\langle PA \rangle$ | - |
| Typhoon-Audio | scb10x/llama-3-typhoon-v1.5-8b-audio-preview | [Model Card] | 8B | $\langle T, A \rangle$, $\langle PA \rangle$ | - |
| Salmonn | tsinghua-ee/SALMONN-7B | [Model Card] | 7B | $\langle T, A \rangle$, $\langle PA \rangle$ | - |
| Glm-4-Voice | THUDM/glm-4-voice-9b | [Model Card] | 9B | $\langle T, A \rangle$, $\langle PA \rangle$ | Interleaving |
| Mini-Omni | gpt-omni/mini-omni | [Model Card] | 0.5B | $\langle T, A \rangle$, $\langle PA \rangle$ | Parallel |
| Mini-Omni2 | gpt-omni/mini-omni2 | [Model Card] | 0.5B | $\langle T, A \rangle$, $\langle PA \rangle$ | Parallel |
| Llama-Omni | ICTNLP/Llama-3.1-8B-Omni | [Model Card] | 8B | $\langle PA \rangle$ | Parallel |
| Audio-Reasoner | zhifeixie/Audio-Reasoner | [Model Card] | 7B | $\langle T, A \rangle$, $\langle PA \rangle$ | - |
| Kimi-Audio | moonshotai/Kimi-Audio-7B-Instruct | [Model Card] | 7B | $\langle T, A \rangle$, $\langle PA \rangle$ | Interleaving and Parallel |
| Qwen2.5-Omni | Qwen/Qwen2.5-Omni-7B | [Model Card] | 7B | $\langle T, A \rangle$, $\langle PA \rangle$ | Interleaving and Parallel |

Table 6: Details of Benchmarking Candidates in this work. Each row lists a tested model's official HuggingFace repository ("Model Card" link), parameter size, supported modalities ("$\langle T, A \rangle$" = Textual Instruction + Audio Input; "$\langle PA \rangle$" = Pure Audio), and whether Interleaving or Parallel multi-modal decoding is applied.

# J CASES

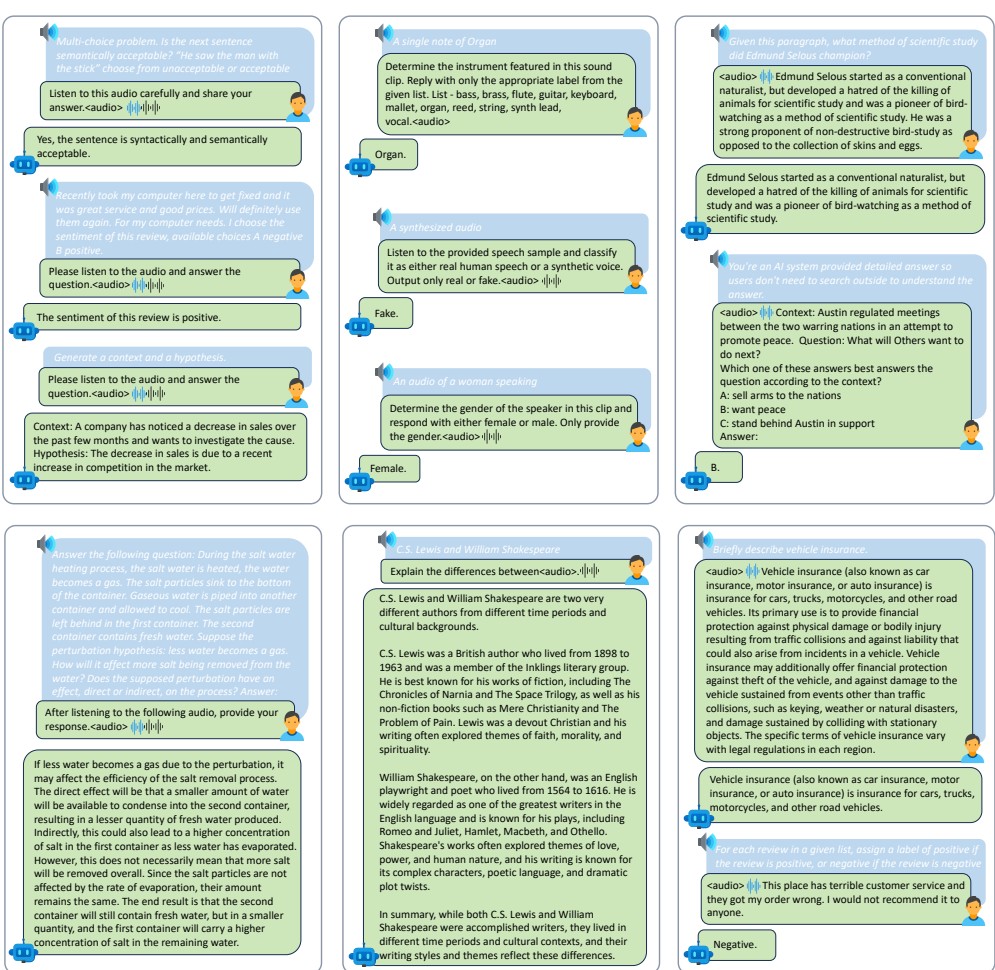

Figure 11: Representative case examples from LLaSO-Base demonstrating the three modality configurations in LLaSO-Eval: pure audio (left), text instruction with audio input (middle), and audio instruction with text input (right). Each column presents distinct tasks under its respective format.

## K  TRAINING DETAILS

### K.1  SYSTEM PROMPT

---

**System Prompt for LLaSO-Base**

A chat between a curious user and an artificial intelligence assistant. The assistant is able to understand the audio content that the user provides, and assist the user with a variety of tasks using natural language. The audio content will be provided with the following format: <Audio>audio content</Audio>.

---

Box 12: Our system prompt for training and evaluation. <Audio>and </Audio>are added into the tokenizer vocabulary as special tokens.

### K.2  PROMPT TEMPLATE

---

**Chat Template for LLaSO-Base**

```
<|begin_of_text|><|start_header_id|>system<|end_header_id|>
```

$X^t_{system-prompt}$ `<|eot_id|><|start_header_id|>user<|end_header_id|>`

$X^*_{query}$ `<|eot_id|><|start_header_id|>assistant<|end_header_id|>`

$X^t_{answer}$ `<|eot_id|>`

---

Box 13: Illustration of the chat template used to construct every training example. We follow the official Llama-3.2 chat template for token ordering and special tokens, while inserting a custom system prompt (full text in Appendix K.1). The user request is encoded as $X^*_{query}$ (see Eq. 1 for the three modality variants), and the model must generate the assistant reply $X^t_{answer}$ followed by the end-of-turn token `<|eot_id|>`. During training, the loss is applied *only* to the assistant's tokens (the last line in this box), teaching the network both the content of the response and where to terminate.

## K.3 TRAINING CONFIGURATION

| Parameter | Stage 1: Modality Alignment | Stage 2: Instruction Tuning |
|---|---|---|
| Device | $4 \times$ NVIDIA A800 | $4 \times$ NVIDIA A800 |
| Model Backbone | Llama-3.2-3B-Instruct | Llama-3.2-3B-Instruct |
| Audio Encoder | Whisper-large-v3 | Whisper-large-v3 |
| Audio Projector | MLP (2-layer, GELU) | MLP (2-layer, GELU) |
| Pretrain Audio Aligner | — | Aligner Checkpoint (from Stage 1) |
| Tune Audio Encoder | False | True/False (optional, see ablation) |
| Tune Audio Projector | True | True |
| Tune LLM | False | True |
| Epochs | 1 | 1 |
| Global Batch Size | 256 | 128 |
| Learning Rate | $1 \times 10^{-3}$ | $3 \times 10^{-5}$ |
| Weight Decay | 0.0 | 0.0 |
| Warmup Ratio | 0.01 | 0.01 |
| LR Scheduler | Cosine | Cosine |
| Max Grad Norm | 1.0 | 1.0 |
| BF16 | True | True |
| Model Max Length | 2048 | 2048 |

Table 7: Training hyperparameters for LLaSO-Base. Stage 1 performs cross-modality alignment, while Stage 2 instruction-tunes the unified model.

## K.4 TRAINING LOSS

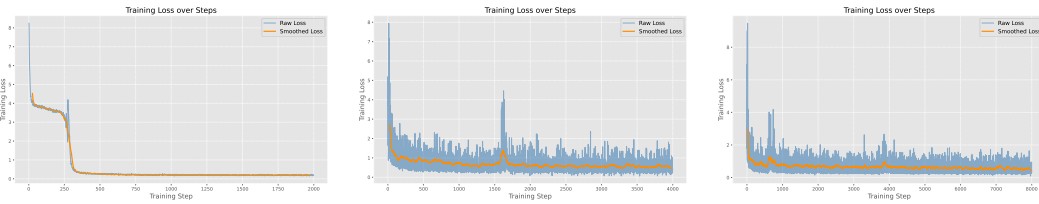

Figure 14: Training loss visualization with Raw Loss and Smoothed Loss. From left to right: (1) alignment stage; (2) instruction tuning stage with frozen encoder; (3) instruction tuning stage with unfrozen encoder.

## L    FOUR STYLES INSTRUCTIONAL PROMPTS

| Prompt Style | Closed-ended Instruction Examples |
|---|---|
| Standardized | Classify the instrument in this audio clip. Choose only from: bass, brass, flute, guitar, keyboard, mallet, organ, reed, string, synth lead, vocal. Output only the label.<audio> |
| Contextualized | For a music classification project, identify the primary instrument in this audio. Return only one of the following: bass, brass, flute, guitar, keyboard, mallet, organ, reed, string, synth lead, vocal.<audio> |
| Stylistic Variation | What is the primary instrument in this audio clip? Respond only with one of: bass, brass, flute, guitar, keyboard, mallet, organ, reed, string, synth lead, or vocal.<audio> |
| Fine-grained Task | Focus only on the instrumental characteristics and determine the correct classification. Output just one label from bass, brass, flute, guitar, keyboard, mallet, organ, reed, string, synth lead, vocal.<audio> |
| **Prompt Style** | **Open-ended Instruction Examples** |
| Standardized | Convert the speech in this audio file into an IPA phonemic sequence. Return phonemes only.<audio> |
| Contextualized | A linguist is analyzing speech samples. Your task is to transcribe the provided audio into an IPA phonemic sequence. Return phonemes only.<audio> |
| Stylistic Variation | Help build a pronunciation guide by converting this audio into IPA phonemes. Return only the phonemes.<audio> |
| Fine-grained Task | Phonetic decoding task: transcribe the provided speech into IPA phonemes and return them without any additional output.<audio> |

Table 8: Representative prompts illustrating the four instruction styles used in our corpus. The closed-ended examples (top) are drawn from the Instrument Classification (IC) task, while the open-ended examples (bottom) are from the Phoneme Recognition (PR) task. Each style, Standardized (direct instructions), Contextualized (scenario-driven), Stylistic Variation (diverse linguistic formulations), and Fine-grained Task (specific sub-aspect focus), is designed to promote compositional generalization across tasks and formats.

# M  MULTI-GRANULARITY SETTING DETAILS

| Coarse-grained (10-year spans) | Medium-grained (5-year spans) | Fine-grained (exact age) |
|---|---|---|
| *Categories: eighties, fifties, forties, nineties, seventies, sixties, teens, thirties, twenties* | *Categories: 15-19, 20-24, 25-29, 30+* | *Range: integer between 18 and 80* |
| Analyze the speaker's voice and determine their age category. Respond only with one of the following: eighties, fifties, fourties, nineties, seventies, sixties, teens, thirties, or twenties.\<audio\> | Based on the audio, identify the speaker's age group. Select one of the following age groups only return: 15-19, 20-24, 25-29, 30+.\<audio\> | Estimate the age of the speaker from the human vocal sounds in this audio clip. Respond with the age only, between 18 and 80.\<audio\> |
| A speech-based recommendation system needs to identify user age. Analyze the voice and classify it into the correct age group from eighties, fifties, fourties, nineties, seventies, sixties, teens, thirties, or twenties.\<audio\> | Can you guess the age group of the speaker in this clip? Please select from the following age groups only return: 15-19, 20-24, 25-29, 30+.\<audio\> | Using this audio, which contains human vocalizations, estimate the speaker's age. Respond with the age as an integer between 18 and 80, no extra information.\<audio\> |
| If the speaker's age appears ambiguous, classify them into the closest matching age group. Select only one label - eighties, fifties, fourties, nineties, seventies, sixties, teens, thirties, or twenties.\<audio\> | Based on the audio, what age group is being used? Pick only return from: 15-19, 20-24, 25-29, 30+.\<audio\> | Listen to this sound sample of human vocalizations and predict the speaker's age as a number between 18 and 80. Provide the age only.\<audio\> |
| Analyze the energy levels, speech rate, and vocal strain in the voice to determine the most accurate age category. Provide only the label from eighties, fifties, fourties, nineties, seventies, sixties, teens, thirties, or twenties. | From the following audio, can you determine the speaker's age group? Options only return: 15-19, 20-24, 25-29, 30+.\<audio\> | Determine the speaker's age based on this recording of human vocalizations. Respond with the age between 18 and 80, without any other explanation. |

Table 9: Some of tasks in our data have granularity. We use Age Classification (AC) task as an examples at three different granularity levels. Coarse-grained prompts elicit classification into decade-based age groups, medium-grained prompts target 5-year age spans, and fine-grained prompts request exact age prediction within a specified range.

# N  Prompts for Pure Audio Modality Format

| Prompt Style | Closed-ended Instruction Examples |
|---|---|
| Standardized | Analyze the provided audio and complete the task mentioned in it.<audio>
Based on the instruction in the audio, provide your response.<audio>
Listen to the audio and respond accordingly.<audio>
Carefully listen to the audio clip and perform the requested action.<audio>
Follow the instruction given in the audio and provide an accurate response.<audio> |
| Contextualized | A voice assistant is asking you to do something. Carefully listen and respond.<audio>
For a comprehension test, listen to the audio and answer the question presented in it.<audio>
In this conversation, the speaker is giving you a directive. Listen and respond appropriately.<audio>
In this experiment, you need to complete the task given in the audio. Provide your response accordingly.<audio>
This is an interactive task. Listen to the speaker and follow their instruction.<audio> |
| Stylistic Variation | Can you understand and complete the request made in this audio?<audio>
If the audio contains a question, answer it accurately. If it contains a command, follow it.<audio>
What action is required in the audio? Complete it and provide your response.<audio>
Make sure to interpret the speaker's request correctly and reply accordingly.<audio>
The speaker in this audio needs a response. Listen and provide a relevant reply.<audio> |
| Fine-grained Task | After hearing the audio, provide your answer to the given task.<audio>
Listen carefully and act according to the instruction in the recording.<audio>
Pay attention to the details in the audio and respond exactly as instructed.<audio>
Understand the content of the audio and give an appropriate response.<audio>
Your task is to carefully analyze the instruction in the audio and execute it properly.<audio> |

Table 10: Examples of text prompts used in the pure-audio modality format, where both the instruction and content are embedded within a single audio stream. The textual cues only instruct the model to listen and respond, without specifying task details. All four prompt styles are included as Table 8 - Standardized, Contextualized, Stylistic Variation, and Fine-grained Task.

## O   EVALUATION TEMPLATE

---

**Instructions:** You are evaluating the performance of an AI assistant in an audio question answering task.

Given a **Reference Answer** and a **Predicted Answer**, assign a score from **1 to 5** based on **Relevance** and **Accuracy**.

**Output Format (exactly, no other text):**

- Score: <integer 1--5>

- Explanation: <concise justification focusing on both relevance and accuracy>

---

**Reference Answer:**

{reference}

**Predicted Answer:**

{predicted}

---

**Please produce the evaluation.**

Figure 15: Evaluation template used for GPT-4o-based scoring of LSLMs' responses. The model assigns an integer score (1–5) according to relevance and accuracy, accompanied by a concise explanation. All results were scored with OpenAI GPT-4o (gpt-4o-mini, Version 2024-07-18).

# P  TASK CATEGORY DEFINITIONS

To facilitate comprehensive and interpretable evaluation, both our training and evaluation datasets are systematically organized into three principal categories: linguistic, semantic, and paralinguistic. This categorization is designed to capture the spectrum of speech-language understanding, from core speech processing and factual reasoning to the nuanced interpretation of speaker traits and acoustic context. Next I will describe the definitions of each categories.

## P.1  LINGUISTIC CATEGORY

Linguistic tasks are aimed at assessing models' basic speech processing ability, primarily through ASR. This foundational category evaluates how accurately a model can transcribe spoken language into text, serving as the backbone for subsequent semantic or paralinguistic inference.

## P.2  SEMANTIC CATEGORY

The semantic category tests a model's ability to extract explicit meaning and perform higher-level reasoning over audio input. In our benchmark, this is represented by the AQA task, which requires models to interpret audio content, combine it with contextual cues, and deliver factual or reasoning-based responses. Although limited to AQA, this category is critical for evaluating the transition from basic perception to comprehension and inference.

## P.3  PARALINGUISTIC CATEGORY

Paralinguistic tasks are structured to probe models' sensitivity to information that lies beyond the literal linguistic content. We further distinguish between speaker-centric and content-centric paralinguistic tasks. Speaker-centric tasks focus on characteristics inherent to the speaker such as gender, age, accent, emotion, and identity capturing traits that are independent of the message being delivered. In contrast, content-centric tasks emphasize cues embedded in the audio signal that reflect content or context, such as phoneme recognition, intent prediction, or entity extraction, irrespective of speaker identity.

## Q DETAILS FOR LLaSO-ALIGN AND LLaSO-INSTRUCT

We have a standardization step in the data construction process as presented in Figure 2. Corrupted or unreadable files are removed, and valid audios are resampled to 16 kHz and stored as WAV/FLAC. Transcripts are filtered to retain only English content with standard characters, then normalized to follow conventional grammar and formatting (e.g., proper capitalization, spacing). Each cleaned sample is paired with a randomly selected instruction template, and the final dataset is packaged in a unified JSON format.

Applying the constructing procedure as Figure 2 yields the finalized training corpora. We provide Table 11 summarizing the resulting task-level composition of LLaSO-Align and LLaSO-Instruct, covering the three categories, linguistic, semantic, and paralinguistic, their 20 sub-tasks, representative data sources, supported input formats, and sample statistics. For the held-out evaluation split, see the stratified breakdown in Table 12.

| Tasks | Descriptions | Data Sources | Modality Formats | Sample Num. | Hours | Instr. Settings |
|-------|-------------|-------------|------------------|-------------|-------|-----------------|
| *Linguistic Task Category* | | | | | | |
| ASR | Automatic Speech Recognition | GigaSpeech
LibriSpeech
LJ Speech
VCTK
MLS | <Textual Instruction, Audio Input>
and
<Audio Instruction, Audio Input> | 12M (LLaSO-Align)&-

1M&0.2M | 47K&-

4K&1K | Open-ended |
| *Semantic Task Category* | | | | | | |
| AQA | Audio Question Answering | Open Orca 1M-GPT4
Open Orca 3.5M-GPT3.5
Stanford Alpaca
Code Alpaca
AlpacaDan
Dolly
OpenOrcaNo
Tigerbot_Alpaca
Tigerbot_Multichat
Unnatural | <Audio Instruction, Audio Input>
and
<Textual Instruction, Audio Input>
and
<Audio Instruction, Textual Input> | 0.4M
0.8M
48K
9K
46K&46K
<1K&3K
7K&10K
20K&20K
6K&33K
0.2M&0.2M | 1.8K
3.6K
<1K
<1K
<1K&<1K
<1K&<1K
<1K&<1K
<1K&<1K
<1K&<1K
<1K&<1K | Open-ended |
| *Paralinguistic Task Category* | | | | | | |
| *Speaker-centric* | | | | | | |
| SGC | Speaker Gender Classification (Biologically) | VoxCeleb1
VCTK
VocalSound
Common Voice | | 35K&35K
71K&71K
20K&20K
0.7M&0.7M | <1K&<1K
<1K&<1K
<1K&<1K
2.3K&1.2K | Closed-ended |
| AC | Accent Classification | VCTK
AccentDB
Common Voice | <Audio Instruction, Audio Input>
and
<Textual Instruction, Audio Input> | 71K&71K
16K&16K
0.3M&0.3M | <1K&<1K
<1K&<1K
2.4K&<1K | Closed-ended and Open-ended |
| AR | Age Recognition (Three Granularities) | VCTK
VocalSound
Common Voice | | 71K&71K
20K&20K
1.2M&1.2M | <1K&<1K
<1K&<1K
5.4K&1.8K | |
| EIE | Emotion Intensity Estimation | MELD
CREMA-D | | 11K&11K
1K&1K | <1K&<1K
<1K&<1K | |
| ER | Emotion Recognition | MELD
CREMA-D | | 9K&9K
7K&7K | <1K&<1K
<1K&<1K | Closed-ended |
| SSD | Synthetic Speech Detection | FoR | | 64K&64K | <1K&<1K | |
| SV | Speaker Verification | MELD | | 11K&11K | <1K&<1K | |
| PSWL | Pronunciation Scoring Word Level | speechocean762 | | 4K&4K | <1K&<1K | Open-ended |
| PSSL | Pronunciation Scoring Sentence Level | | | 4K&4K | <1K&<1K | |
| *Content-centric* | | | | | | |
| PR | Phoneme Recognition | Phonemizer Generated | | 1M&1M | 5K&4K | |
| SCR | Speech Command Recognition | Speech Commands | | 68K&68K | <1K&<1K | |
| IP | Intent Prediction | SLURP | <Audio Instruction, Audio Input>
and
<Textual Instruction, Audio Input> | 71K&71K | <1K&<1K | Open-ended |
| EE | Entity Extraction | | | 45K&45K | <1K&<1K | |
| VSC | Vocal Sound Classification | VocalSound | | 20K&20K | <1K&<1K | |
| IC | Instrument Classification | NSynth | | 0.2M&0.2M | 1.1K&0.2M | Closed-ended |
| ISC | Instrument Source Classification | | | 0.3M&0.3M | <1K&<1K | |
| PP | Pitch Prediction | | | 0.3M&0.3M | <1K&<1K | Open-ended |
| VC | Velocity Classification | | | 0.3M&0.3M | 1.1K&<1K | Closed-ended |
| **Total** | - | - | - | ~25.5M | ~89.5K | - |

Table 11: Overview of task-level composition in LLaSO-Align and LLaSO-Instruct, spanning three core categories, linguistic, semantic, and paralinguistic, across 20 sub-tasks. Each entry summarizes representative data sources, supported input formats, and sample-level statistics. LLaSO-Eval is constructed as a stratified evaluation set presented in Table 12.

# R    DETAILS FOR LLaSO-EVAL

To complete our data trio, we introduce LLaSO-Eval, a held-out evaluation suite designed to accompany the LLaSO training set. Derived from the same underlying corpus but separate from the training split, LLaSO-Eval covers 15,044 samples across 20 tasks, categorized into linguistic, semantic, and paralinguistic categories. Moreover, it supports all three major modality configurations and tests both within- and cross-modal generalization. We provide task breakdown in Table 12.

From a task perspective, LLaSO-Eval enables comprehensive evaluation of model capabilities across three major categories: linguistic, semantic, and paralinguistic tasks. Within the paralinguistic category, tasks are further distinguished into into speaker-centric (e.g., gender, age, accent) and content-centric (e.g., intent prediction, phoneme recognition). This distinction enables fine-grained analysis of how models handle both speaker identity and acoustic-semantic information. From a modality perspective, by supporting three major configurations, LLaSO-Eval not only evaluates model performance on seen modalities configures but also tresses cross-modal generalization, testing robustness to novel input combinations. Additionally, to evaluate instruction-following capabilities, LLaSO-Eval includes both open-ended tasks for free-form comprehension and reasoning, and closed-ended tasks requiring predefined label selection. This allows for quantitative measurement of instruction adherence through metrics such as abstention rate.

| Tasks | Descriptions | Data Sources | Modality Formats | Sample Num. | Metrics |
|---|---|---|---|---|---|
| *Linguistic Task Category* | | | | | |
| ASR | Automatic Speech Recognition | GigaSpeech
LibriSpeech
LJ Speech
VCTK
MLS | <Textual Instruction,
Audio Input > | 4566 | WER&CER |
| *Semantic Task Category* | | | | | |
| AQA | Audio Question Answering | Open Orca 1M-GPT4 Mukherjee et al. (2023)
Open Orca 3.5M-GPT3.5
Stanford Alpaca Taori et al. (2023)
Code Alpaca Chaudhary (2023)
AlpacaDan Jordan (2023)
Dolly Conover et al. (2023)
OpenOrcaNo RuterNorway (2023)
Tigerbot_Alpaca Research (2023)
Tigerbot_Multichat Chen et al. (2023)
Unnatural Honovich et al. (2022) | <Pure Audio >

and

<Audio Instruction,
Textual Input > | 100
100
100
100
100&100
100&100
100&100
100&100
100&100
100&100 | GPT-4o |
| *Paralinguistic Task Category* | | | | | |
| *Speaker-centric* | | | | | |
| SGC | Speaker Gender Classification (Biologically) | VoxCeleb1 Nagrani et al. (2017)
VCTK
VocalSound Gong et al. (2022)
Common Voice Ardila et al. (2019) | | 100&100
100&100
200&200
100&100 | ACC |
| AC | Accent Classification | VCTK
AccentDB Ahamad et al. (2020)
Common Voice | | 100&100
100&100
100&100 | ACC |
| AR | Age Recognition (Three Granularities) | VCTK
VocalSound
Common Voice | <Pure Audio >
and
<Textual Instruction,
Audio Input > | 100&100
200&200
100&100 | ACC/MAE |
| EIE | Emotion Intensity Estimation | MELD Poria et al. (2018)
CREMA-D Cao et al. (2014) | | 100&100
100&100 | ACC |
| ER | Emotion Recognition | MELD
CREMA-D | | 100&100
100&100 | ACC |
| SSD | Synthetic Speech Detection | FoR Reimao & Tzerpos (2019) | | 100&100 | ACC |
| SV | Speaker Verification | MELD | | 100&100 | ACC |
| PSWL | Pronunciation Scoring Word Level | speechocean762 Zhang et al. (2021) | | 200&200 | GPT-4o |
| PSSL | Pronunciation Scoring Sentence Level | | | 200&200 | ACC&GPT-4o |
| *Content-centric* | | | | | |
| PR | Phoneme Recognition | Phonemizer Generated  Bernard & Titeux (2021) | | 100&100 | PER |
| SCR | Speech Command Recognition | Speech Commands Warden (1804) | | 100&100 | WER&CER&GPT-4o |
| IP | Intent Prediction | SLURP Bastianelli et al. (2020) | <Pure Audio >
and
<Textual Instruction,
Audio Input > | 858&858 | GPT-4o |
| EE | Entity Extraction | | | 569&569 | GPT-4o |
| VSC | Vocal Sound Classification | VocalSound | | 200&200 | ACC |
| IC | Instrument Classification | | | 100&100 | ACC |
| ISC | Instrument Source Classification | NSynth Engel et al. (2017) | | 100&100 | ACC |
| PP | Pitch Prediction | | | 112&112 | MAE |
| VC | Velocity Classification | | | 100&100 | ACC |
| **Total** | - | - | - | 15044 | - |

Table 12: Overview of LLaSO-Eval composition. This stratified evaluation set, sampled from LLaSO-Instruct, includes 20 tasks across linguistic, semantic, and paralinguistic categories (sub-divided into speaker-centric and content-centric). For each task, we provide data sources, modality formats, sample counts, and evaluation metrics. Automatic metrics are used where applicable, with GPT-4o-based judgment for open-ended tasks.

## S  BASELINE PERFORMANCE DETAILS

| Task | Qwen2-Audio | Typhoon-Audio | Salmonn | Glm-4-Voice | Mini-Omni | Mini-Omni2 | Llama-Omni | Audio-Reasoner | Kimi-Audio | Qwen2.5-Omni | LLaSO-Base | LLaSO-Base (Unfrozen) | Metrics |
|---|---|---|---|---|---|---|---|---|---|---|---|---|---|
| ASR | 0.22 | 0.11 | 0.86 | 0.93 | 0.95 | 0.95 | 0.88 | 0.28 | 0.14 | 0.40 | **0.08** | 0.14 | WER↓ |
|  | 0.12 | 0.06 | 0.69 | 0.79 | 0.81 | 0.80 | 0.73 | 0.12 | 0.05 | 0.26 | **0.03** | 0.07 | CER↓ |
| AQA | 2.41 | 1.76 | 1.47 | 2.22 | 1.42 | 1.57 | 1.97 | 2.44 | **2.94** | **2.94** | 2.06 | 2.27 |  |
|  | 2.42 | 1.77 | 1.41 | 2.34 | 1.47 | 1.53 | 2.02 | 2.24 | 2.70 | **3.09** | 1.80 | 2.27 |  |
|  | 2.73 | 2.16 | 1.41 | **3.29** | 1.75 | 2.05 | 2.99 | 2.51 | 3.22 | 3.22 | 2.39 | 2.23 |  |
|  | 2.78 | 2.22 | 1.72 | 2.93 | 1.45 | 1.51 | 2.48 | 2.86 | **3.45** | 2.63 | 1.46 | 1.98 | GPT-4o↑ |
|  | 2.56\|3.47 | 1.87\|2.69 | 2.05\|2.04 | 2.49\|3.09 | 1.63\|1.42 | 1.66\|1.68 | 2.38\|2.73 | 2.22\|2.84 | **3.28\|3.69** | 2.99\|3.46 | 2.57\|2.72 | 2.54\|3.22 |  |
|  | 3.49\|3.62 | 3.14\|2.91 | 3.13\|3.03 | 3.21\|**4.06** | 1.54\|1.32 | 1.64\|1.50 | 2.95\|3.78 | 3.42\|3.95 | 3.77\|4.01 | **3.80\|3.88** | 2.48\|2.62 | 2.42\|2.84 |  |
|  | 2.13\|3.29 | 1.61\|2.47 | 1.42\|2.42 | 2.51\|1.68 | 1.22\|1.17 | 1.26\|1.41 | 1.88\|2.29 | 2.12\|2.88 | **3.35\|3.38** | 3.20\|**3.58** | 1.71\|2.28 | 1.96\|2.47 |  |
|  | 3.14\|1.29 | 2.83\|1.68 | 2.96\|1.83 | 3.11\|1.03 | 2.34\|1.21 | 2.52\|1.29 | 3.16\|1.11 | 3.07\|1.54 | **3.53\|1.16** | 2.96\|1.19 | 2.74\|2.23 | 2.80\|**2.43** |  |
|  | 3.13\|3.14 | 3.04\|3.04 | 3.12\|3.19 | 2.82\|3.10 | 1.33\|1.27 | 1.42\|1.31 | 2.72\|3.08 | 2.91\|3.13 | **3.38\|3.16** | 3.19\|3.15 | 3.05\|**3.74** | 3.09\|3.68 |  |
|  | 2.20\|2.52 | 2.36\|1.91 | 2.37\|1.58 | 1.97\|1.98 | 1.41\|1.20 | 1.43\|1.28 | 2.20\|2.09 | 2.14\|2.09 | 2.71\|2.77 | 2.12\|2.42 | **2.90**\|2.60 | 2.87\|**3.26** |  |
| SGC | **1.00** | 0.85 | 0.59 | 0.11 | 0.14 | 0.11 | 0.36 | 0.38 | 0.98 | 0.53 | 0.96 | 0.88 |  |
|  | 0.95 | 0.77 | 0.44 | 0.12 | 0.00 | 0.00 | 0.26 | 0.32 | 0.97 | 0.41 | **0.99** | **0.99** | ACC↑ |
|  | 0.67 | 0.59 | 0.13 | 0.04 | 0.00 | 0.02 | 0.03 | 0.38 | 0.66 | 0.40 | **0.76** | 0.61 |  |
|  | **0.99** | 0.67 | 0.18 | 0.07 | 0.00 | 0.00 | 0.26 | 0.37 | 0.81 | 0.35 | 0.91 | 0.97 |  |
| AC | 0.16 | 0.21 | 0.22 | 0.07 | 0.00 | 0.00 | 0.07 | 0.14 | 0.38 | 0.06 | **0.52** | 0.40 |  |
|  | 0.12 | 0.14 | 0.32 | 0.09 | 0.06 | 0.03 | 0.14 | 0.18 | 0.31 | 0.19 | 0.83 | **0.86** | ACC↑ |
|  | 0.05 | 0.11 | 0.10 | 0.03 | 0.00 | 0.00 | 0.07 | 0.02 | 0.20 | 0.02 | 0.73 | **0.78** |  |
| AR | 0.23 | 0.10 | 0.26 | 0.02 | 0.00 | 0.00 | 0.16 | 0.23 | 0.17 | 0.11 | **0.70** | 0.68 | ACC↑ |
|  | **0.52** | 0.12 | 0.06 | 0.01 | 0.00 | 0.00 | 0.17 | 0.12 | 0.12 | 0.08 | 0.50 | 0.38 |  |
|  | 18.69 | 20.47 | 11.24 | 15.35 | 21.34 | 18.46 | *Reject* | 13.57 | 12.07 | 10.31 | 10.32 | **8.78** | MAE↓ |
| EIE | 0.54 | 0.40 | 0.31 | 0.13 | 0.04 | 0.03 | 0.31 | 0.52 | **0.65** | 0.52 | 0.48 | 0.45 | ACC↑ |
|  | 0.31 | 0.24 | 0.24 | 0.08 | 0.04 | 0.06 | 0.08 | 0.29 | 0.34 | 0.27 | **0.48** | 0.37 |  |
| ER | 0.24 | 0.28 | 0.30 | 0.14 | 0.04 | 0.03 | 0.16 | 0.32 | **0.52** | 0.29 | 0.17 | 0.16 | ACC↑ |
|  | 0.30 | 0.12 | 0.21 | 0.02 | 0.07 | 0.01 | 0.04 | 0.28 | 0.32 | **0.33** | 0.26 | 0.30 |  |
| SSD | 0.43 | 0.46 | 0.50 | 0.10 | 0.11 | 0.12 | 0.26 | 0.35 | 0.63 | 0.42 | **0.99** | **0.99** | ACC↑ |
| SV | 0.25 | 0.20 | 0.19 | 0.04 | 0.03 | 0.03 | 0.08 | 0.16 | 0.22 | 0.15 | **0.32** | 0.16 | ACC↑ |
| PR | 1.19 | 3.08 | 1.82 | 0.90 | 0.92 | 0.97 | 1.46 | 1.08 | 1.58 | 1.28 | **0.03** | **0.03** | PER↓ |
| SCR | 2.60 | 0.98 | 1.09 | 1.00 | 1.00 | 1.00 | 20.19 | 2.40 | 1.00 | 3.53 | **0.04** | 0.05 | WER↓ |
|  | 2.40 | 0.85 | 0.75 | 0.98 | 0.98 | 0.97 | 21.11 | 1.84 | 0.31 | 3.52 | **0.02** | 0.04 | CER↓ |
|  | 3.04 | 3.13 | 4.07 | 1.85 | 1.39 | 1.89 | 1.58 | 4.10 | 4.56 | 3.91 | **4.86** | 4.80 | GPT-4o↑ |
| IP | 2.52 | 1.86 | 1.88 | 1.78 | 1.26 | 1.26 | 1.80 | 2.29 | 2.05 | 2.00 | **3.93** | 3.90 | GPT-4o↑ |
| EE | 2.73 | 2.85 | 3.29 | 2.34 | 1.42 | 1.60 | 2.34 | **3.78** | 3.57 | 2.78 | 3.57 | 3.44 | GPT-4o↑ |
| PSWL | 1.86 | 2.04 | 1.32 | 1.62 | 1.24 | 1.16 | 1.28 | 2.61 | **3.30** | 1.25 | 2.90 | 2.68 | GPT-4o↑ |
| PSSL | 0.17 | 0.33 | 0.13 | 0.24 | 0.00 | 0.00 | 0.13 | 0.09 | 0.20 | 0.27 | **0.39** | 0.24 | ACC↑ |
|  | 1.95 | 1.71 | 1.38 | 1.84 | 1.46 | 1.54 | 1.38 | 2.03 | 2.76 | 2.12 | **2.80** | 2.66 | GPT-4o↑ |
| VSC | 0.85 | 0.49 | 0.61 | 0.32 | 0.02 | 0.03 | 0.03 | 0.59 | 0.84 | **0.92** | 0.78 | 0.82 | ACC↑ |
| IC | **0.60** | 0.16 | 0.16 | 0.00 | 0.01 | 0.01 | 0.00 | 0.20 | 0.26 | 0.51 | 0.50 | 0.55 | ACC↑ |
| ISC | 0.60 | 0.16 | 0.16 | 0.03 | 0.06 | 0.03 | 0.12 | 0.17 | 0.38 | 0.44 | 0.60 | **0.77** | ACC↑ |
| PP | 19.02 | 36.83 | 41.92 | 40.20 | 61.49 | 59.32 | *Reject* | 32.68 | 31.64 | 18.37 | **8.02** | 10.55 | MAE↓ |
| VC | 0.02 | 0.17 | **0.22** | 0.08 | 0.00 | 0.00 | 0.07 | 0.15 | 0.19 | 0.12 | 0.18 | 0.20 | ACC↑ |

Table 13: Comprehensive evaluation across multiple LSLMs in LLaSO-Eval. Blue highlights denote best performance per task. *Reject* indicates that, during manual inspection, for 95% or more of the responses in the corresponding open-ended setting/task, the model explicitly expresses inability to assist or process the task, states it is a text-only model unable to recognize audio, or behaves as a pure text model by asking the user to describe the audio, its content, or information therein. From SGC to VC we only tested the <Textual Instruction, Audio Input>format, because for these tasks we also used only the <Textual Instruction, Audio Input>format data of those tasks during training.

## T DISCLOSURE OF LLM ASSISTANCE

We used LLM-based assistants **only** to aid and polish writing. Assistance was limited to grammar and style edits such as tightening wording, improving flow and transitions, shortening captions, harmonizing terminology, and light LaTeX phrasing. All technical content, claims, experimental design, data processing, modeling, analysis, figures/tables, and conclusions were conceived and produced by the authors; LLMs did not contribute novel ideas, code, datasets, evaluations, or result interpretations and are not contributing authors.

