# OpenReview forum: "LLaSO: A Reproducible Foundation for Large Speech-Language Models"
_ICLR.cc/2026/Conference — ICLR 2026 Conference Withdrawn Submission_

### Official Review · Reviewer_4yJd · 2025-10-26

**Soundness:** 3
**Presentation:** 2
**Contribution:** 3
**Rating:** 4
**Confidence:** 4

**Summary:**

In this work, the authors open source a large-scale speech–language modeling system including model, training configuration, training and evaluation dataset. It is helpful for the community for the future research work on the spoken language model. My main concern is clarity. There are a lot of essential information missing, especially on the dataset construction.
It makes other researchers hard to adapt this baseline giving the missing information. Detailed questions are listed in the questions section.

**Strengths:**

- An open source large-scale speech–language modeling system is provided.
- Many efforts have been done to make the ASR datasets good for modality alignment task, such as introducing hand-crafted instruction templates to convert the ASR datasets for speech language model alignment.
- Many efforts have been done on the instruct dataset, which includes linguistic tasks, semantic tasks, and paralinguistic tasks.
- An evaluation dataset is provided
- A baseline model is provided with detailed analysis

**Weaknesses:**

- Some essential dataset construction information is missing
- The dataset doesn't include multi-turn dialogue dataset

**Questions:**

- LLASO-ALIGN.
  -  ``18 hand-crafted instruction templates that frame the task with varying specificity and constraints''. Why 18 instruction templates? What aspects are they addressed? What can be improved? Is there any stasts for training samples with different instruction templates?
  -  Figure 2 and Appendix Q describe dataset standardization. What exactly have been achieved compared with the original datasets?
- LLASO-Instruct.
  -  ``for each task, we manually construct 20 text instructions across four prompt styles''. Why every task has 20 text instructions, nothing more or less? What's the main motivation/goal for four prompt styles? What's the stats for the training data with those instructions?
- LLASO-Eval:
   - How do we compute the normalized score across different tasks?
- How do you define open-ended/closed ended instruct settings in Table 11?
- ``LLaSO-Eval covers 15,044 samples across 20 tasks''. Is this big enough?
- The comparison in Table 13 is questionable. For example, the WER for other models are too high and many of them are above 50\%. Is this expected?

**Details Of Ethics Concerns:**

A dedicated ethics section is provided.

---

### Official Review · Reviewer_GTZ1 · 2025-11-01

**Soundness:** 2
**Presentation:** 2
**Contribution:** 2
**Rating:** 2
**Confidence:** 4

**Summary:**

This paper introduces LLaSO, a fully open framework for large speech-language models that integrates data, evaluation, and a reference model for transparency and reproducibility. It provides an alignment corpus and a multi-task instruction set that covers linguistic, semantic, and paralinguistic tasks, along with a benchmark supporting multiple input and instruction modalities. A reference model trained on these public resources serves as a baseline, showing that broader task coverage improves performance, though significant gaps remain in pure-audio and cross-modality generalization.

**Strengths:**

- The paper aims to integrate training (Align + Instruct) and evaluation within a single, open framework that supports multiple modality configurations, helping reduce the entry barrier for large speech-language model research.
- Having 40% of tasks to paralinguistics is a positive shift from the purely semantic objectives that have dominated many previous datasets.
- The analysis of abstention rates and modality stability offers useful insights into model robustness across different input types.

**Weaknesses:**

- The novelty is very limited. The model design, which projects audio into the LLM space followed by alignment and supervised fine-tuning, has become standard in recent speech language model research.
- The authors claim that prior instruction datasets are "primarily semantic with limited modalities," but several existing works already include style or paralinguistic coverage (e.g., SIFT-50M, Audio-FLAN).
- Using GPT-4o for evaluation raises reproducibility concerns. Even with the temperature fixed at 0, GPT-4o outputs can still vary due to hidden randomness or version updates.
- It is unclear how GPT-4o behaves as an evaluator, as there are no statistics showing its alignment with human judgment in the reported experiments.
- Treating ASR as the sole "speech-text alignment" objective is not well motivated for paralinguistic learning, because ASR focuses on transcript accuracy and may ignore or discard speaker and prosodic information.
- In the paper, the authors mention "semantic tasks are already well-represented." However, to the best of my knowledge, evaluations on existing benchmarks show speech LMs still lag behind ASR+LLM pipelines on several SLU tasks.
- The related work section provides very limited discussion. Since the paper aims to present a unified framework for training data, model architecture, and benchmarking, the authors should include more detailed comparisons of each component with prior work.

**Questions:**

Please refer to my comments in the weakness section. I also have a few additional comments and questions below:

- Why are duration (hours) missing for AVQA, COTA, OpenAQA, OpenASQA, and SIFT‑50M in Table 1?
- Some task-dataset pairings do not seem reasonable. For example, using MELD for speaker verification is questionable since VoxCeleb is typically used for that task.
- The proposed evaluation benchmark appears to overlap with Dynamic-SUPERB. Several tasks and datasets are already covered there, and both frameworks share a similar design that includes abstention rate analysis for certain tasks. I would appreciate if the authors can justify why a new benchmark is necessary instead of extending existing open ones. Although this work introduces more modality combinations (e.g., audio instruction with text input), its coverage for text instruction with audio input remains noticeably behind existing benchmarks.

---

### Official Review · Reviewer_UjN1 · 2025-11-03

**Soundness:** 2
**Presentation:** 2
**Contribution:** 2
**Rating:** 2
**Confidence:** 4

**Summary:**

This paper introduces LLaSO, an open, end-to-end stack for Large Speech-Language Models (LSLMs) consisting of: 1. LLaSO-Align (12M instruction-formatted ASR pairs), 2. LLaSO-Instruct (13.5M multi-task instruction-tuning samples over 20 tasks), and 3. LLaSO-Eval (15,044 stratified evaluation samples).

The authors also release LLaSO-Base, a 3.8B parameter reference system (Whisper-large-v3 encoder + 2-layer MLP projector + Llama-3.2-3B-Instruct) trained on public data only; it attains a min-max normalized overall score of 0.72 on LLaSO-Eval, exceeding the next best model’s 0.65. The work argues that broader task coverage reduces abstention and improves generalization, while speech-only setups remain the hardest.

**Strengths:**

The key strength of this paper is open and end-to-end release.

**Weaknesses:**

English-only (for now). The authors acknowledge this as a limitation; given the positioning as a “foundation,” multilingual extensions should be higher priority or at least partially demonstrated (e.g., a pilot split).

The evaluation is dependent on GPT-4o. Having some human eval or cross check from different LLM would be better.

Synthetic data make scientific study harder, e.g. how's different TTS model affect the quality, scaling etc.

No scaling behavior been studied.

Lack of architecture study and no generation.

**Questions:**

As a foundational open model for LLM, scaling behavior and different architecture study are important. Given the paper is still hook up different model, I feel more experimental results with different size and open model are necessary to support the foundation claim.

---

### Note · Authors · 2025-11-15

I have read and agree with the venue's withdrawal policy on behalf of myself and my co-authors.